# Off to new Shores: A Dataset & Benchmark for (near-)coastal Flood Inundation Forecasting

**Brandon Victor**
La Trobe University
b.victor@latrobe.edu.au

**Mathilde Letard**
University of Rennes
mathilde.letard@univ-rennes.fr

**Peter Naylor**
ESA Φ-lab
peter.naylor@esa.int

**Karim Douch**
ESA Science Hub
karim.douch@esa.int

**Nicolas Longépé**
ESA Φ-lab
nicolas.longepe@esa.int

**Zhen He**
La Trobe University
z.he@latrobe.edu.au

**Patrick Ebel**
ESA Φ-lab
patrick.ebel@esa.int

## Abstract

Floods are among the most common and devastating natural hazards, imposing immense costs on our society and economy due to their disastrous consequences. Recent progress in weather prediction and spaceborne flood mapping demonstrated the feasibility of anticipating extreme events and reliably detecting their catastrophic effects afterwards. However, these efforts are rarely linked to one another and there is a critical lack of datasets and benchmarks to enable the direct forecasting of flood extent. To resolve this issue, we curate a novel dataset enabling a timely prediction of flood extent. Furthermore, we provide a representative evaluation of state-of-the-art methods, structured into two benchmark tracks for forecasting flood inundation maps i) in general and ii) focused on coastal regions. Altogether, our dataset and benchmark provide a comprehensive platform for evaluating flood forecasts, enabling future solutions for this critical challenge. Data, code & models are shared at `https://github.com/Multihuntr/GFF` under a CC0 license.

## 1 Introduction

Floods are among the most impactful natural disasters, both in terms of the societal as well as the economic costs they impose (15). The consent amongst climate scientists and disaster relief experts is that this trend will aggravate in the coming decades(50; 36; 27; 61), close by rivers (21) and in particular near the coastlines due to rising sea levels and more severe extreme weather events (35; 10; 45; 65; 64; 46). Yet, closeness to waterways is of economical importance such that endangered regions have grown in population, thus bringing more people at the risk of floods (66; 60).

Hence, international collaborations and efforts e.g. in the context of the *United Nations (UN) Sustainable Development Goals (SDG)* (62; 58) tackle climate change mitigation and adaptation. According to the *Early warnings for all* initiative of the World Meteorological Organization and the UN, every person on Earth shall be protected by early warning systems until 2027 (67), but to date significantly more effort is required towards covering the Global South and developing early warning systems for coastal inundation (71). In line with these needs, our contribution is a global dataset and benchmark for a timely forecasting of flood extent maps. Our novel Global Flood Forecasting (GFF) dataset represents climate zone and continent distributions of events as reported in the Dartmouth

38th Conference on Neural Information Processing Systems (NeurIPS 2024) Track on Datasets and Benchmarks.

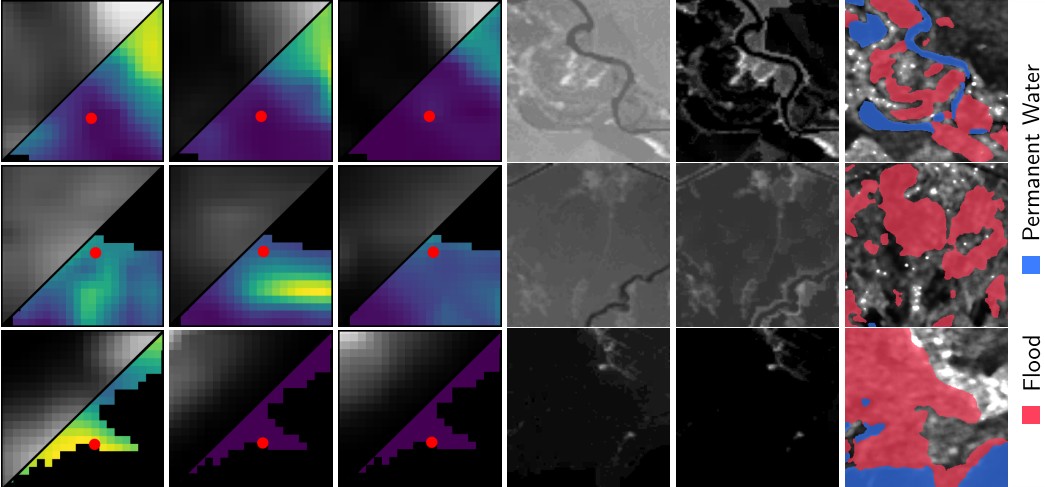

Figure 1: **Exemplary data.** Columns: Three ERA5 and ERA5-Land time series samples, DEM, Height Above Nearest Drainage (HAND). Pre-flood Sentinel-1 (S1) overlaid with target-time flood map. Columns 1-3 provide context data at a coarse scale, red dots indicates the coverage of columns 4-6 at fine scale. Rows: Three examples, showcasing floods near a river, settlement and coastline. The events are due to heavy rain, tropical storms and a storm surge, illustrating the diversity of GFF.

Flood Observatory (DFO) (66), while focusing on (near-)coastal areas and their varied drivers of flood hazards. To underline the diversity of cases covered by GFF, Fig. 1 illustrates examples of flood due to heavy rain, plus high tides in the last case. Each sample combines multi-temporal atmospheric reanalysis products, high resolution terrain models and simulated precipitation drainage, hydrological basin attributes, as well as pre-flood Sentinel-1 (S1) radar satellite observations and flood extent annotations as targets. While offering vast information, combining this multi-modal and multi-scale data poses technical challenges, especially for hand-crafted solutions common in flood forecasting.

Meanwhile, data-driven machine learning has been a major cause of recent breakthroughs in modeling of ungauged rivers (38; 53) and rapid flood mapping (11; 66; 12). While the former may help anticipating river run-off and the latter can support ongoing relief efforts, there's a lack of research on ahead-of-time prediction of the inundation maps themselves and the comparability of such forecasting models on a common benchmark dataset. This is unfortunate, as the timely availability of flood extent maps would allow humanitarian agencies to undertake preparatory measures such as the evacuation of endangered population ahead of time rather than acting post-hoc. Though forecasting of inundation maps provides critical information for disaster preparedness, few prior works tackle this challenge and there is an absence of benchmarks to facilitate such developments (51). The aim of our work is to fill this gap by introducing GFF as a global dataset and benchmark for flood extent forecasting, analysis-ready for modern machine learning approaches. In sum, our main contributions are:

- The curation of GFF, a novel global multi-modal multi-temporal dataset for (near-)coastal flood forecasting, derived from six distinct sensors and products at two separate scales of resolution. Through careful stratification, we sample regions representative of climate zones and continents for which major floods have been recorded.

- The design of two tracks for i) general flood extent prediction and ii) with a focus on separating coastal versus near-coastal and inland floods. Each region experiences distinct climate, weather and flood drivers such that tailored approaches may be most fruitful.

- The benchmarking of established methods, to provide the reader with a comprehensive overview of the diverse landscape of methods and the state-of-the-art on the defined tracks.

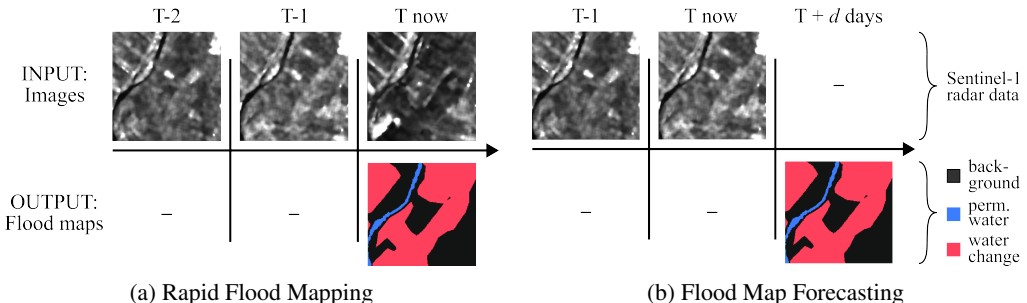

(a) Rapid Flood Mapping         (b) Flood Map Forecasting

Figure 2: **Conceptual similarities and differences** between the two tasks of a) in-event rapid flood mapping and b) pre-event forecasting potential flood maps. While the former focuses on detecting change of water coverage in observations at prior dates versus now, the latter is about predicting such change at a given lead time $d$ where no observation is yet available. Beyond internalizing the physical signatures of moisture and water, this requires learning the dynamics of associated flood drivers.

## 2 Related Work

### 2.1 River streamflow & runoff forecasting

The close relation between riverine floods and a river's weather-driven run-off sustain general interests in river streamflow modeling. Classical forecasting approaches are simulation-based and build on complex, hand-crafted features calibrated to encode basin-specific properties and dynamics (3; 8). More recent data-driven models are based on deep neural networks and demonstrated the ability to forecast streamflow at ungaged sites (38; 53). The limitations of such approaches are that they lack any direct relation to actionable flood extent maps, as there is no straightforward translation from river run-off to actual inundation maps. The closest efforts pursuing this endeavor are given by the prior work of (54) who relate river run-off to historical flood extent observed via multispectral satellites by querying records in a lookup table. The shortcomings of this approach are its reliance on optical imagery which may be affected by clouds, and especially the reliance on historic flood event observations at every region of interest. Furthermore, the intermediate modeling of run-off oftentimes either relies on access to river gauges or focuses on upstream regions nearby the source, which renders it inapplicable to coastal areas. In contrast to this work and prior efforts, our dataset seeks to enable research and development for a timely and all-weather forecasting of flood inundation maps on a global scale, not constrained to sites where historic flood observations are readily available for and including (near-)coastal regions.

### 2.2 Rapid mapping

The spaceborne mapping of floods and their impacts in order to support disaster relief is a central mandate of several international services and charters (9; 2). While rapid mapping and its automation have a long history (22; 14; 26), recent progress in machine learning offers access to products at a pace and accuracy outmatching expert hand-labeled annotations (48; 12). Key to this is the availability of large-scale annotated datasets (47; 66; 12). Our work builds on the recent Kuro Siwo dataset of S1 observations and models (12) to obtain reference flood maps on par in terms of quality with post-event expert annotations (12), subsequently post-processed and used as forecasting targets herein. The task tackled in our work is related to rapid mapping, but demands the timely forecasting of inundation maps and thus allows for pre-disaster preparatory measures. Accordingly, the setup of available observations and target date of predictions differs across both tasks, as conceptualized in Fig. 2. While rapid mapping is about the detection of physical properties such as surface soil moisture and water mass, our forecasting task requires translating atmospheric dynamics and their hydro-meteorological impact onto land surface while taking into account factors like local topography and basin properties. Independent of their similarities and differences, forecasting and rapid mapping are complementary in their function and both are crucial for disaster mitigation and relief, respectively.

| Dataset | Task | Sample size | Resolution | Sample count | Static input | Dynamic input | Event count | Timestamps |
|---|---|---|---|---|---|---|---|---|
| SEN12-FLOOD (59) | classification | 512 × 512 | 10 m | 336 | - | S1, S2 | 3 | circa 9-14 |
| OMBRIA (18) | segmentation | 256 × 256 | 10 m | 1,688 | - | S1, S2 | 23 | 1 Pre + Post |
| S1GFloods (63) | segmentation | 256 × 256 | 10 m | 5,360 | - | S1 | 46 | 1 Pre + Post |
| CAU-Flood (31) | segmentation | 256 × 256 | 10 m | 18,302 | - | S1, S2 | 18 | 1 Pre + Post |
| Kuro Siwo (12) | segmentation | 224 × 224 | 10 m | 67,490 | DEM | S1 | 43 | 2 Pre + Post |
| GRDC GRDB (13) | regression | 1D sequence | in-situ | 10,000+ | - | river gauges | - | 10,000+ |
| HYSETS (6) | regression | 1D sequence | in-situ, basin & 10 - 30 km | 14,425 | basin properties | river gauges, NRCan + SCDNA + Livneh + ERA5(-Land) | - | 10,000+ |
| Caravan (39) | regression | 1D sequence | in-situ & basin | 10,000+ | HydroATLAS | river gauges, ERA5-Land | - | 10,000+ |
| **Global Flood Forecasting (ours)** | **segmentation** | **224 × 224** | **10 m & 5 - 30 km** | **163,873** | **DEM, HAND, HydroATLAS** | **S1, GloFAS, ERA5(-Land)** | **298** | **20 Pre + Post** |

Table 1: **Overview of datasets** for flood mapping (top) and flood forecasting (bottom) purposes. The former feature high resolution imaging while the latter focus on in-situ time series. GFF enables forecasting of flood extent by curating sequences of gridded products at high spatial resolution.

# 3 Data

The GFF dataset focuses on (near-)coastal regions characterized by a diversity of causes underpinning flood hazard—ranging from pluvial, fluvial or coastal drivers such as storm surges, to potential compound events. The resulting dataset includes 298 Regions of Interest (ROI) experiencing an equal count of spatially and temporally separated flood events in the years 2014-2020. For each flood event, the dataset contains flood drivers as input and flood segmentation maps at event time as targets. The dataset is accompanied by a pre-defined 5-fold cross-validation benchmark split to enable fair comparison between models and drive innovation. All data are provided in a rasterized *TIFF* file format (72) and prepared to facilitate developing and evaluating data-driven machine learning models. Beyond the flood maps, observations and products provided herein, the GFF dataset is extendable and comes with all scripts needed to expand to further regions, flood events or to include new modalities.

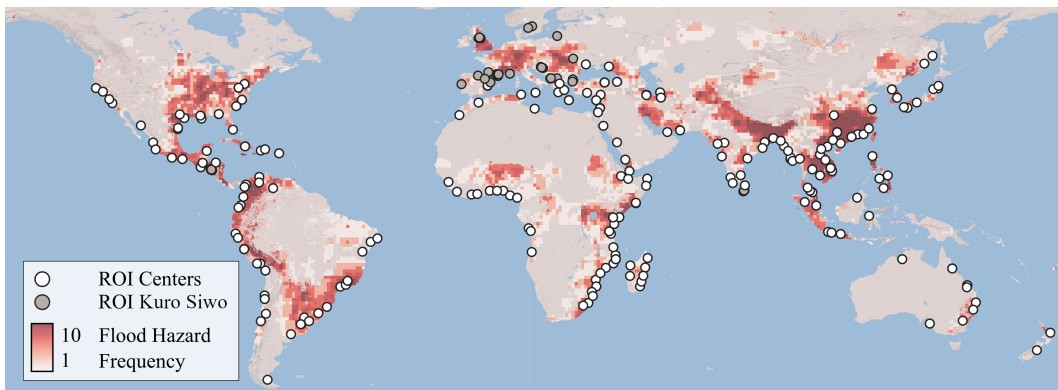

Figure 3: **Map** of dataset. Points are centers of curated ROI and their associated flood events. Red shadings indicate the distribution of global flood hazard frequency (15). Many of the endangered regions are close to the sea, especially in the (sub-)tropics. Unlike most prior work addressing well-monitored or upstream regions, these areas are particularly well represented in our dataset.

## 3.1 Identifying candidate Regions Of Interest

To collect a diverse set of floods on a global scale, we cross-reference the exhaustive records of DFO with HydroBASIN level 8 basins (40) underlying events featured in three prior works: the recent Kuro Siwo dataset for rapid mapping (12), a list of 280 landfalling tropical cyclones which caused extreme precipitation and subsequent flooding (68), and finally any (near-)coastal basins characterized by HydroATLAS (42; 41)—a global compendium of river basins and their attributes.

*First*, all of Kuro Siwo's expert-labelled data in the years of our analysis are included for training and testing. While its distribution spans six continents, its focus is on Europe in the well-monitored northern extratropical lattitudes. Complementary, the *second* group of basins feature inland level 8 basins at the epicenters of cyclones' land footprints, which mostly cover (sub-)tropical areas. Finally,

the *third* group of basins are those indicated by HydroATLAS to be at or near the coast, filling in any properties not yet sufficiently captured by the preceding two sources of events.

To prioritize on the subset of most relevant ROI and focus on hazardous events, candidate areas of all three sources are sorted by impact and affected population as given by DFO and HydroATLAS. All candidate ROI and dates are iterated through and checked for coincidence with S1 orbits. For creating global flood extent labels at the desired quality, at least one S1 image during the flood time period and two S1 pre-event images are required. Floodmaps were not generated for a ROI within $< 5$ °proximity of a previously selected ROI or if its climate zone is already represented sufficiently.

## 3.2 Capturing the diverse drivers of flood & flood extent forecasting

Each sample in the dataset consists of multi-modal inputs as shown in Fig. 1, paired with flood-map segmentation labels. The various input types are split into two scales: ERA5, ERA5-Land, HydroATLAS and downsampled CopDEM30 are provided as a *coarse context*. Pre-event Sentinel-1, CopDEM30 and HAND are given at the fine *local scale* of the floodmaps. Most forcings are static, but at the coarse scale ERA and ERA5-Land are multi-temporal. In summary, the inputs of GFF are:

- **ERA5** and **ERA5-Land** (32; 52), containing daily atmospheric state reanalysis. The provided $9 + 14$ atmospheric variables are corresponding to the forcings used in established data-driven river-runoff models (37; 54; 53). ERA5 and ERA5-Land complement each other: the first provides vital information of weather conditions over both the land and sea. The latter focuses on land at a significantly finer horizontal resolution for better forcings.

- **HydroATLAS** (42), providing static multi-level basin attributes including a subset of 87 hydro-environmental key properties like 'aridity index' and 'average wet season' as selected by stream-flow experts (37; 54; 53). HydroATLAS is licensed to be freely available for any research project, and is provided in this dataset as raster data.

- **Sentinel-1** (S1) Ground Range Detected (GRD) Synthetic Aperture Radar images (25), preprocessed via ESA's SNAP toolbox (76) and replicating the pipeline of Kuro Siwo (12). In the temporal proximity of excessive rainfall, cloud-penetrating radar measurements are more useful than optical imagery. Observations at the closest time *before* the event's recorded onset are used as input, while images *during* the event are used for the label generation.

- **Copernicus Digital Elevation Model** at global 30 meters resolution (CopDEM30) (1) and **Height Above Nearest Drainage** (HAND) maps (7) derived from the surface model via NASA's HydroSAR package (49). The former provides a representation of the ROI's terrain, while the latter is a downstream product specifying the local topology's drainage potentials.

- Daily aggregated **river discharge and runoff water** history modeled via the Global Flood Awareness System (GloFas) (29), part of the operational flood forecasting within the Copernicus Emergency Management Service. Explicit, hydrological modeling of such variables is a key step in operational flood forecasting systems (4; 17; 54). To explore the forecasting of extent maps, we include this modality in our dataset and encourage investigating models with and without it for exploring one versus two-stage modeling.

Conceptually, in an initial step the contextual information allows a model to integrate weather, soil and elevation data over the surrounding topography for a time window starting 20 days before the target date. In a second stage, the model can then use the processed contextual representation together with finer resolved information to fill in details of exactly where the water will go in the local area. The data are preprocessed as detailed in the corresponding section of the accompanying *Datasheet*.

## 3.3 Generating floodmaps anywhere

A central design objective of GFF is to cover a diverse set of continents, climate zones and land cover types beyond ROI which are already monitored and served well. For this sake, floodmaps are computed by ensembling two rapid mapping models pretrained on the Kuro Siwo dataset and further refined in a post-processing step. The two utilized models are a masked auto-encoder pre-trained ViT (30; 16) and a SNUNet change detector (20). Both models use pre- and in-event Sentinel-1 images to classify whether a pixel of the in-event Sentinel-1 image became flooded or not. For rapid mapping,

the models classify no-water pixels with F1 scores of 99.07 and 98.97 and water pixels with F1 scores of 87.58 and 86.52 on hold-out data, respectively. A third set of floodmaps is generated by ensembling the logits of both models. All three types of floodmaps are released with the GFF dataset, but the ensemble labels are considered the default. The exception are ROI with hand-annotated expert labels collected in Kuro Siwo, which are utilized herein upon availability.

For creating labels at a given ROI, a uniform grid is created over the scene and tiles of $224 \times 224$ pixels size which intersect with HydroRIVER geometries are added to an initial search set. The tiles in this set are sorted following the upstream flow of any adjacent river and the 200 most downstream cells are selected. Orthogonal to the grid-covered river stream, a buffer of 2 additional tiles are added to both sides of this set, resulting in up to $200 \times (1 + 2 \times 2)$ initial tiles per ROI. Starting from these at most 1000 tiles, a conditional floodfill algorithm is used to search for affected areas. Specifically, floodmaps are generated for each tiles and if any tile shows significant flooding (defined as $> 5\%$ of the tile), then a buffer of 3 neighboring tiles is added to expand the open set of search tiles. This process terminates at a maximum of 2500 tiles per ROI or when no more flooding is encountered.

### 3.4 Label post-processing

The models utilized for generating labels achieve high performance in the rapid mapping scenario (12), but their outputs were found to exhibit artifacts when deployed in practice. These include tiling artifacts and speckled segmentation artifacts at multiple levels, which we corrected for as follows.

First, as is common for neural networks operating on individual local tiles one at a time, the generated outputs initially displayed serious tiling artifacts. These artifacts are cleaned for by generating logits of 50%-overlapping tiles and then taking a weighted average at each pixel. The resulting averaged logits smoothly transition between adjacent tiles. This method entirely removes all tiling artifacts.

Second, the initial maps displayed strong speckling patterns induced by the upconvolution operations used in the pre-trained models (55). We used a $5 \times 5$ pixels majority filter to determine a more reliable boundary between classes and applied a contour-finding algorithm to remove any remaining blobs smaller than 50 pixels (43; 56). Importantly, these post-processing steps were not performed on a tile-by-tile basis but on the full area after combining tiles.

Moreover, it was found during preliminary testing that the pretrained SNUNet model (20) performed better when driven by CopDEM30 compared to the DEM originally used in (12), potentially due to CopDEM30's better quality, and thus we use CopDEM30 for generating floodmaps with SNUNet.

Finally, to improve the delineation between flood and permanent water pixels in the ensembled generated maps, we merge both classes into the former and superimpose the permanent water labels from ESA WorldCover (73) for the latter. This was found to improve separation of both water classes.

### 3.5 Selecting representative ROI via stratification

Following the selection process of section 3.1, candidate sites are proactively filtered to ensure a representative distribution of prior flood events across continents and climate zones. To obtain a reference, we estimate the true distribution of worldwide flood events using the Cartesian product of DFO events and level 8 basins. Each flood $\times$ basin pairing is counted for the basin's continent/climate zone. An expected distribution is created by scaling down all bins in the true distribution such that we expect to produce floodmaps for 2000 level 8 basins. While iterating over all potential ROI, a ROI is skipped if its climate zone is already well represented. The distributions and ROI iterations are calculated within each continent separately to allow for continents with different proportions of climate zones affected by floods, and ensure that the dataset has global coverage.

Stratification is complicated by the fact that any ROI's floodmap may extend to multiple level 8 basins but which ones are covered isn't known prior to rapid mapping. Therefore, a ROI is skipped if over half of its area is covered by climate zones which are already sufficiently well represented. While generating, a ROI may or may not show flooding. As ROI without flooding are less relevant than ROI with flooding for the creation of our dataset, they were included but counted at half for the statistics. Flooded ROI may vary in their extent if DFO indicates long time intervals. In this case, multiple S1 passes may be available and we pick the timestamp exhibiting the largest flood extent.

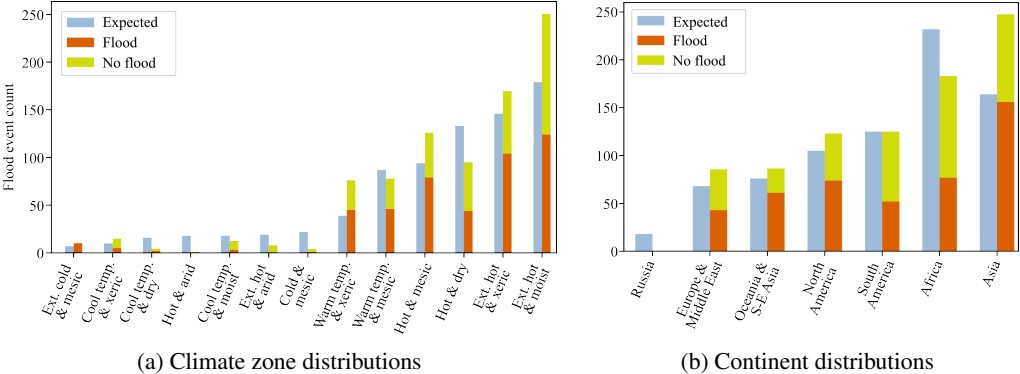

(a) Climate zone distributions       (b) Continent distributions

Figure 4: **Empirical distributions of floods** in our dataset regarding their frequency across different a) climate zones and b) continents. In both regards, the histogram of cumulative flood (orange) and no flood observations (green) qualitatively mirrors the expected occurrence of all global flood events reported by the DFO (blue), with minor discrepancies due to the (near-)coastal focus.

The outlined procedure results in 298 ROI of flood events depicted in the map of Fig. 3 (99 of which are listed by both DFO and the database of (68), indicating cyclone-driven floods). The selected regions cover 1049 level 8 sub-basins and a total of 164845 tiles with 9.5 % of the tiles featuring flood extent, for all of which input forcings and segmentation labels as described in sections 3.2 and 3.3 are provided. The histograms of climate zone and continent coverage are depicted in Fig. 4 and confirm that the distribution of global flood history recorded by the DFO is well represented.

## 3.6 Data splits

The GFF dataset is accompanied by a suggested split into five partitions for a 5-fold cross-validation setup. Partitions are defined to not leak any test information into optimization. This is accomplished by partitioning the whole world by HydroATLAS level 4 basins, ensuring hydrologically well-defined groupings. ROI are assigned to the partition of largest overlap and adjacency is avoided by excluding ROI closer than circa 500 km to one another during the candidate selection stage. Finally, the defined splits are backward-compatible to river gauge splits of the popular Caravan (39) dataset for river streamflow modeling, allowing the community to explore synergies between both contributions.

## 4 Benchmarks

To highlight the value of the GFF dataset, we benchmark a diverse and representative set of state-of-the-art models on two distinct tracks: global flood extent forecasting and a separate focus on coastal floods. In both cases, the task is to predict a flood segmentation map of binary values ($B$: background vs $W$: water) at a given lead time $d$ days after the pre-flood Sentinel-1 observation, with $d$ varying per flood event. For evaluating the performance of flood extent forecasts, we mask the permanent water pixels given by ESA WorldCover labels as described in section 3.4 and then evaluate the F1 score of the binary forecasts over all other pixels, checking whether a pixel became flooded or not. This is to focus on the pixels potentially undergoing class change rather than on permanent water bodies whose dynamics are comparably static. We report overall F1 score and scores for both classes individually.

For every track, the experimental setup is a 5-fold cross-validation scheme, with one partition as the test set, another as the validation set and the remaining three for training. For each baseline, all F1 scores are reported as the means of performances of model instances across partitions, plus their cross-fold standard deviation in F1 scores in order to provide an estimate of cross-run spread.

The first track measures the performance of forecasting inundation maps across all ROI included. The second track evaluates models specifically on ROI separated into *coastal* versus *near-/non-coastal* areas, with tiles categorized whether they are less or more than 10 km distant from the nearest coast. This distinction is to differentiate between areas directly impacted by coastal hazards compared to regions only affected by near-shore weather dynamics, which may require differences in modeling.

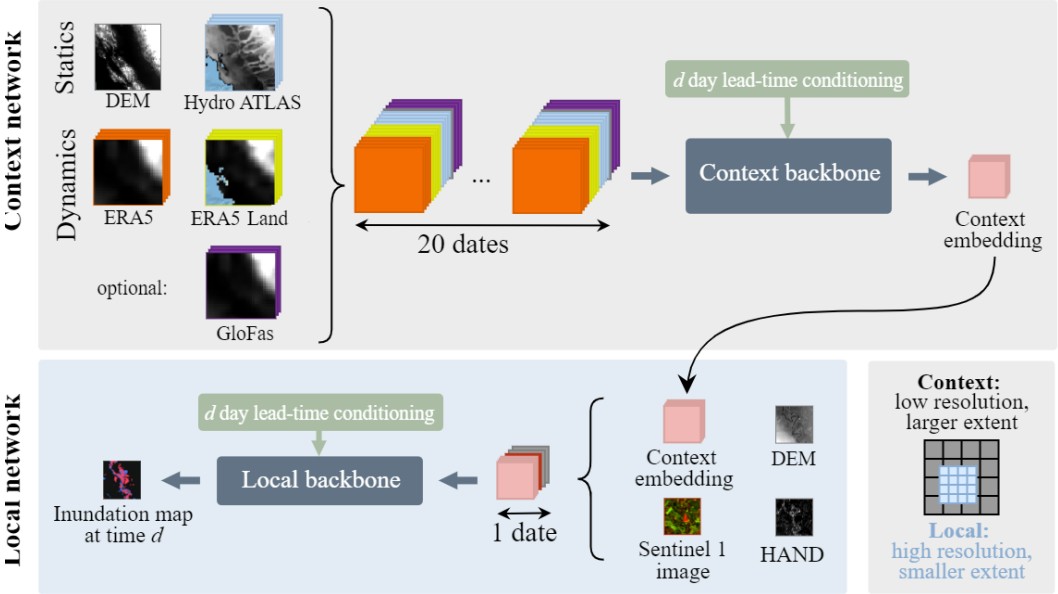

Figure 5: **Baseline model platform design,** accommodating for i) a *context network (top)* which processes a spatio-temporal sequence of coarse resolution context data and whose output feature embeddings are then processed by ii) a *local network (bottom)*, concatenated with local high resolution data. The two network backbones (in dark grey) are placeholders for the different baselines benchmarked herein. The final output is a flood segmentation forecast at a given lead time.

## 4.1 Baselines

To provide baselines for our benchmark, we propose a novel two-level pipeline which serves as a platform to combine existing state-of-the-art spatio-temporal architectures with minimal extras.

The platform is designed to process input forcings of coarse context and local information. The horizontal resolution of context forcings such as GloFas is at $0.05°$ (circa $5.5$ km at the equator), but local information such as S1 images and the flood maps are resolved to a $10$ m resolution, finer by three orders of magnitude. To our knowledge there are no existing solutions processing spatial information at such difference in scale. Although there exist works that utilise both S1 and ERA5 at the same time (69), they do not do so while modelling spatial relationships. This challenge invites for innovative technical contributions from the broader machine learning audience.

Per setup, our platform hosts any of the diverse established baselines—ranging from convolutional models, LSTM (44), to temporal attention (23) and vision transformers (70; 5). In all cases, i) the context inputs (ERA5, ERA5-Land, HydroATLAS & CopDEM) are embedded using any of the aforementioned multi-temporal architectures. Then, ii) a local segmentation model takes the aggregated context embedding along with the local data (Sentinel-1, CopDEM30 & HAND) to predict a floodmap, conditioned on a given lead time of $d$ days via Feature-wise Linear Modulation (FiLM) (57). For each setup incorporating a specific baseline, the context and the local modules share the same backbone architecture, but do not share weights. The described pipeline is illustrated in Fig. 5. As a fifth baseline, we implement a simple logistic regression model (33; 24). Analogous to the prior work of (24), we train it on the mono-temporal local information and report the learned channel weightings to provide interpretation of feature importance in the Supplementary Material.

Each baseline is evaluated following the 5-fold cross-validation protocol, trained via the ADAM optimizer (34) at a batch size of $16$ for $10$ epochs with an initial learning rate of $10^{-3}$ and an exponential decay of $0.8$. The cost function is a binary cross entropy loss with class weightings of $0.5$ and $5$ for background versus water classes — roughly equal to each category's inverse frequency. Using the same loss, models are evaluated on the validation partition every epoch and the checkpoint with best validation loss is used for testing. Computations have been performed on-premises at ESA, training on NVIDIA RTX A6000 GPU. All code and checkpoints are made public.

## 4.2 Track 1: Global flood map prediction

This track is on translating all contextual spatio-temporal and local high-resolution data detailed in section 3.2 to flood extent forecasts at a given lead time, as described in the preceding paragraphs.

The results are reported in Table 2, whose left half shows the overall and class-wise F1 scores for each of the five baselines. U-TAE performs best overall, closely followed by LSTM U-Net. Interestingly, the strength of LSTM-based models for flood extent forecasting mirrors the method's competitiveness in the related task of river streamflow forecasting (38), for which it likewise is the state-of-the-art (37). Complementary to these outcomes, the right side of the table reports F1 scores when only evaluating on the hand-labeled data of Kuro Siwo (KS). Compared to the previous results, there is a trend of performance decrease but most differences are within a standard deviation, which are generally higher on the Kuro Siwo labels. Even though it now is LSTM U-Net obtaining best results, we conclude that performances are roughly comparable for annotated and our derived labels. In summary, the best baselines can forecast flood inundation at an F1 score of circa 0.77, with strong performances on the background class. However, the prediction of flooded pixels is significantly more challenging, as this is the minority class and may benefit most from future research and modeling.

Table 2: **Track 1.** Benchmarking of five baselines on the GFF dataset. U-TAE performs best overall, while LSTM U-Net is best when only evaluating on the subset of KS hand-annotated ROI.

| Model | F1 | F1-B | F1-W | $F1_{KS}$ | $F1\text{-}B_{KS}$ | $F1\text{-}W_{KS}$ |
|---|---|---|---|---|---|---|
| U-TAE (23) | $\mathbf{0.77 \pm 0.04}$ | $\mathbf{0.97 \pm 0.00}$ | $\mathbf{0.57 \pm 0.07}$ | $0.72 \pm 0.02$ | $0.97 \pm 0.01$ | $0.48 \pm 0.05$ |
| LSTM U-Net (44) | $0.76 \pm 0.04$ | $\mathbf{0.97 \pm 0.00}$ | $0.55 \pm 0.08$ | $\mathbf{0.73 \pm 0.03}$ | $0.97 \pm 0.01$ | $\mathbf{0.50 \pm 0.07}$ |
| 3DConv U-Net (44) | $0.76 \pm 0.04$ | $0.97 \pm 0.01$ | $0.54 \pm 0.08$ | $0.70 \pm 0.08$ | $0.97 \pm 0.02$ | $0.43 \pm 0.16$ |
| MaxViT U-Net (70; 5) | $0.75 \pm 0.03$ | $0.96 \pm 0.01$ | $0.53 \pm 0.06$ | $0.73 \pm 0.05$ | $\mathbf{0.98 \pm 0.01}$ | $0.49 \pm 0.09$ |
| logistic regression (33) | $0.66 \pm 0.04$ | $0.93 \pm 0.02$ | $0.40 \pm 0.07$ | $0.65 \pm 0.10$ | $0.94 \pm 0.04$ | $0.36 \pm 0.16$ |
| U-TAE GloFAS (23) | $0.76 \pm 0.05$ | $\mathbf{0.97 \pm 0.00}$ | $0.55 \pm 0.08$ | $0.73 \pm 0.05$ | $0.97 \pm 0.01$ | $0.49 \pm 0.10$ |

## 4.3 Track 2: Coastal versus near-coastal & inland flood map prediction

The focus of this track is on disentangling the distinct challenges of forecasting coastal versus near-coastal floods, separately analyzed and benchmarked herein. (Near-)coastal floods are difficult to model for established approaches not only due to their potential compound nature (74), but also due to being most distant from the headwater basin at which river-runoff models typically excel (38; 39). To provide further insight, we follow the preceding experimental protocol but evaluate the baselines distinctly on the dataset's subset of coastal versus near-coastal and inland areas. Regions 10 km or closer to the nearest shore are considered as coastal, a distance about the size of one ERA5-Land cell.

The outcomes are shown in Table 3, the left and right sides contain outcomes for coastal versus near-coastal and inland areas, respectively. Overall, baseline performances are higher at coastal regions. This may be due to the incidental presence of coastal wetlands whose re-occurring inundation are more predictable, or due to the high count of cyclone-driven floods in the dataset whose extreme precipitation is most destructive near the coasts even more so than right at the shores (28; 19; 75). Overall, the regional separation reveals performance differences, which deserve further research.

Table 3: **Track 2.** Benchmarking of five baselines on the GFF dataset, separating (*c*)oastal versus (*n*)ear-coastal & inland areas. U-TAE and 3DConv U-Net perform best at the coasts, while U-TAE performs best on inland regions. Altogether, baseline performances are higher at coastal regions.

| Model | $F1_c$ | $F1\text{-}B_c$ | $F1\text{-}W_c$ | $F1_n$ | $F1\text{-}B_n$ | $F1\text{-}W_n$ |
|---|---|---|---|---|---|---|
| U-TAE (23) | $\mathbf{0.80 \pm 0.06}$ | $\mathbf{0.95 \pm 0.02}$ | $0.65 \pm 0.11$ | $\mathbf{0.76 \pm 0.03}$ | $\mathbf{0.98 \pm 0.00}$ | $\mathbf{0.55 \pm 0.07}$ |
| LSTM U-Net (44) | $0.78 \pm 0.07$ | $0.94 \pm 0.03$ | $0.63 \pm 0.11$ | $0.75 \pm 0.04$ | $0.97 \pm 0.00$ | $0.53 \pm 0.07$ |
| 3DConv U-Net (44) | $0.80 \pm 0.08$ | $\mathbf{0.95 \pm 0.02}$ | $\mathbf{0.65 \pm 0.09}$ | $0.74 \pm 0.04$ | $0.97 \pm 0.01$ | $0.50 \pm 0.09$ |
| MaxViT U-Net (70; 5) | $0.78 \pm 0.06$ | $0.94 \pm 0.03$ | $0.63 \pm 0.10$ | $0.74 \pm 0.03$ | $0.97 \pm 0.01$ | $0.50 \pm 0.06$ |
| logistic regression (33) | $0.73 \pm 0.04$ | $0.92 \pm 0.02$ | $0.54 \pm 0.06$ | $0.65 \pm 0.05$ | $0.93 \pm 0.02$ | $0.36 \pm 0.09$ |
| U-TAE GloFAS (23) | $0.78 \pm 0.08$ | $0.95 \pm 0.02$ | $0.62 \pm 0.10$ | $0.75 \pm 0.04$ | $0.97 \pm 0.0$ | $0.52 \pm 0.09$ |

# 5  Discussion

**Potential societal impact:** Our work and the future research it may facilitate closely align with the UN *Sustainable Development Goals* (62; 58). We do not foresee any direct adversarial impact of our work on society, and all information utilized herein is already made publicly accessible by the responsible authorities (e.g. ASF & NASA, Copernicus, ECMWF, ESA) in line with their mandates.

**Known limitations & future work:** The definition of what is considered a flood is contested in the context of ecosystems such as salt marshes, swamps and practices like irrigation farming. Rather than focusing on such non-hazardous scenarios, disasters listed by the DFO and close to populated areas as indicated via WorldPop guide our dataset curation. However, non-hazardous and semi-persistent land submersion may still be included in our dataset in case they occur in a harmful event's periphery.

The post-event mapping algorithm, although we build upon the state-of-the-art and apply further post-processing, is not flawless. While being more sophisticated than established heuristic approaches, our exchange with operational flood forecasting teams clarified that a separate and regularly re-running of permanent water detection is oftentimes employed in practice. Accordingly, we recommend a combination of our approach with such a complementary step to meet the best forecasting practices.

Finally, the meteorological forcings included in the dataset are reanalysis rather than reforecasts. This is to disentangle challenges in the quickly advancing field of weather forecasting from the task at the heart of our work, so benchmarking outcomes can be isolatedly attributed to the latter. However, using reanalysis as forcings may overestimate downstream task performances compared to utilizing forecasts during operational deployment. Hence, we recommend evaluating developed models using forecasts as forcings in case they shall be deployed in practice.

# 6  Conclusions

To tackle the challenge of timely flood extent forecasting and empower researchers to address this matter on a global scale, we curate the first dataset and public benchmark on this task. The dataset features multi-modal and multi-temporal data of (near-)coastal flood events distributed across six continents and 13 climate zones, encompassing a variety of events driven by pluvial, fluvial, coastal or compound hazards. As diverse as these cases, the collected observations and products pose unique technical challenges inviting for innovative contributions from the scientific community—ranging from optimal solutions for sensor fusion to multi-scale Earth observation modeling. While we consider these technical challenges stimulating for a broader machine learning audience, it is the societal importance of disaster risk reduction via timely forecasting that we wish to highlight at last.

## Acknowledgments and Disclosure of Funding

We thank SmartSat CRC for funding the research visit of Brandon Victor at ESA Φ-lab. Furthermore, we would like to thank our colleagues at ESA, as well as Frederik Kratzert and Adi Gerzi Rosenthal at Google Research's Flood Forecasting Department for the fruitful discussions and precious feedback.

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
