# Supplementary Material and Datasheet:

## Off to new Shores: A Dataset & Benchmark for (near-)coastal Flood Inundation Forecasting

## Contents

## 1 Introduction

This supplementary document follows the Datasheets for Datasets template of (8) to document the Global Flood Forecasting (GFF) dataset and its creation.

Further resources are provided:

- in the accompanying publication https://arxiv.org/abs/2409.18591

- in the GitHub repository https://github.com/Multihuntr/GFF

- on the Zenodo platform https://zenodo.org/records/13133267

# 2 Datasheets for Datasets

| **Motivation** |
| --- |

**For what purpose was the dataset created?** Was there a specific task in mind? Was there a specific gap that needed to be filled? Please provide a description.

Our dataset was created for the research and development of flood extent forecasting approaches. Existing datasets only either focus on river stream-flow prediction (but don't translate run-off into flood extent) (15; 16) or rapid mapping of satellite-acquired (5). Furthermore, existing datasets are oftentimes biased towards well-monitored sites in the Global North as well as basins close to the source or head basin of rivers. This focus excludes (near-)coastal areas, especially in regions which would benefit most from an improvement of early warning systems, such as Developing Countries and islands. Our contribution addresses this gap by providing a multi-modal dataset and benchmark for timely forecasting of flood extent, with a carefully stratified representation of continents and climate zones.

**Who created this dataset (e.g., which team, research group) and on behalf of which entity (e.g., company, institution, organization)?**

The dataset was created at the European Space Agency's Φ-Lab by

- Brandon Victor, who is a PhD student at La Trobe University and was a visiting researcher at ESA Φ-Lab.

- Mathilde Letard, who is post-doc at University of Rennes and was a visiting researcher at ESA Φ-Lab.

- Peter Naylor, who is staff at ESA Φ-Lab.

- Karim Douch, who is staff at ESA Science Hub.

- Nicolas Longepe, who is staff at ESA Φ-Lab.

- Zhen He, who is professor at La Trobe University

- Patrick Ebel, who is staff at ESA Φ-Lab.

**Who funded the creation of the dataset?** If there is an associated grant, please provide the name of the grantor and the grant name and number.

The dataset creation received no dedicated funding, but the research visits of Brandon Victor and Mathilde Letard at the European Space Agency's Φ-Lab was funded by their respective home institutes as indicated in the authors' affiliations.

**Any other comments?**

We would like to thank our colleagues at ESA, as well as Frederik Kratzert and Adi Gerzi Rosenthal at Google Research for the fruitful discussions and precious feedback.

| Composition |
|:-:|

**What do the instances that comprise the dataset represent (e.g., documents, photos, people, countries)?** Are there multiple types of instances (e.g., movies, users, and ratings; people and interactions between them; nodes and edges)? Please provide a description.

Each instance in the dataset pertains to rasterized geospatial information, which either captures the atmospheric state or the land surface. More specifically, each instance represents a small section of a flood extent map and the associated driving factors leading up to it.

**How many instances are there in total (of each type, if appropriate)?**

There are a total of 164845 tiles of size $224 \times 224$ pixels, spread across 298 sites which are distributed amongst six continents and 13 climate zones.

**Does the dataset contain all possible instances or is it a sample (not necessarily random) of instances from a larger set?** If the dataset is a sample, then what is the larger set? Is the sample representative of the larger set (e.g., geographic coverage)? If so, please describe how this representativeness was validated/verified. If it is not representative of the larger set, please describe why not (e.g., to cover a more diverse range of instances, because instances were withheld or unavailable).

The dataset assembles co-registered and stratified data of a subset of the following global observations and products: ERA5, ERA5-Land, GloFAS, HydroAT-LAS, CopDEM30, HAND and Sentinel-1 (S1) images. The dataset contains 298 sites with input forcings at 20 different dates during flood events in the years 2014-2021. The global distribution of flood events per climate zone was identified by combining the Dartmouth Flood Observatory (DFO) flood archive (23) and HydroATLAS (18; 17). Subsequently, sites and times were chosen to match this global distribution. The inclusion of a site was also dependent on S1 image availability, thus it is not perfectly representative. In particular, some climate zones with rarer flood events are slightly underrepresented.

**What data does each instance consist of? "Raw" data (e.g., unprocessed text or images) or features?** In either case, please provide a description.

Each instance in this dataset is a mapping from various raster data to an output segmentation map. The input samples in the dataset consists of multi-modal forcings. All input types are split into two scales: ERA5, ERA5-Land,GloFAS, HydroATLAS and downsampled CopDEM30 are provided as a *coarse context* at 0.05° (roughly 5.5 km resolution at the equator); and pre-event Sentinel-1, CopDEM30 and HAND are provided at the 10 m resolution *local scale* of the floodmaps. Most of these sources are static, but at the coarse scale ERA, ERA5-Land and GloFAS are multi-temporal. In sum, the inputs in GFF are:

- **ERA5** and **ERA5-Land** (12; 20), containing daily atmospheric state reanalysis.The provided atmospheric variables are corresponding to the forcings used in established data-driven river-runoff models (14; 22; 21).

ERA5 and ERA5-Land complement each other, as the first provides vital information of extreme weather conditions over both the land and ocean while the latter focuses on weather over land at a significantly finer horizontal resolution to provide better forcings.

- **HydroATLAS** (18), providing static multi-level basin attributes including a subset of 87 hydro-environmental key properties like 'aridity index' and 'average wet season' as selected by stream-flow experts (14; 22; 21). HydroATLAS is licensed under a Creative Commons Attribution (CC-BY) 4.0 International License to be freely available for any research project, and is provided in this dataset as raster data.

- **Sentinel-1** (S1) Ground Range Detected (GRD) Synthetic Aperture Radar images (10), cessed via ESA's Sentinel Processing Toolbox (SNAP) (26) and replicating the pipeline of Kuro Siwo (5). In the temporal proximity of excessive rainfall, cloud-penetrating radar measurements are more useful than optical imagery.

- the global **Copernicus Digital Elevation Model** at 30 meters resolution (CopDEM30) (1) and **Height Above Nearest Drainage (HAND) maps (4)** derived from the surface model via NASA's HydroSAR package. CopDEM30 provides a representation of the ROI's terrain, while HAND is a downstream product which specifies the local topology's drainage potentials.

- Daily aggregated **river discharge and runoff water** history modeled via the Global Flood Awareness System (GloFAS) (11), part of the operational flood forecasting within the Copernicus Emergency Management Service. Explicit, hydrological modeling of such variables is a key step in operational flood forecasting systems (2; 6; 22). To facilitate the forecasting of extent maps, we include this modality in our dataset and encourage investigating models with and without it for exploring one versus two-stage modeling.

The output segmentation maps consist of three classes: background, permanent water, flood and are obtained and preprocessed as detailed in sections *Collection Process* and *Preprocessing/cleaning/labeling*. Altogether, Fig. 1 illustrates the different data sources, and how they constitute the input and target data.

**Is there a label or target associated with each instance?** If so, please provide a description.

There is a label associated with each instance. It is a floodmap with three classes: background, permanent water and flooded pixels. It is generated with a model trained on the Kuro Siwo (5) rapid flood mapping dataset and then post-processed to remove artifacts.

**Is any information missing from individual instances?** If so, please provide a description, explaining why this information is missing (e.g., because it was

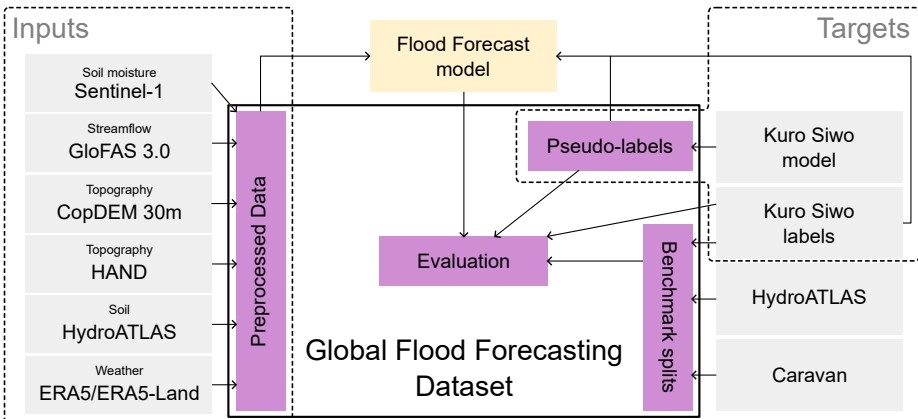

Figure 1: **Schematic overview** of the various sources of data that constitute the inputs and target of our global flood forecasting dataset.

unavailable). This does not include intentionally removed information, but might include, e.g., redacted text.

The inputs for each input-output pair can contain NaN values where the source dataset does not have any data. For example, each input contains a wide context, sometimes including the ocean. However, ERA5-Land and CopDEM30 are specifically pertaining to land surface and thus are undefined on the ocean.

**Are relationships between individual instances made explicit (e.g., users' movie ratings, social network links)?** If so, please describe how these relationships are made explicit.

Contrary to most tiled datasets, the usage pattern of this dataset is to read small tiles from a larger file on the fly. Thus, individual tiles are grouped by site into a single raster file. The dataset includes a shapefile to describe the coordinates of each tile within the site. This means that all of the tiles at each site is easy to visualise in GIS software, such as QGIS.

**Are there recommended data splits (e.g., training, development/validation, testing)?** If so, please provide a description of these splits, explaining the rationale behind them.

The dataset prescribes five partitions which are to be used for cross-validation. These partitions were created by randomly partitioning the world by HydroATLAS's level 4 basin geometries. Then each site was assigned to one of the partitions based on which HydroATLAS level 4 basin it overlapped with the most. This method was chosen to make this dataset backward-compatible with related global datasets. For example, if one wants to train a model for the streamflow forecasting task via the Caravan dataset (16) and subsequently utilize those predictions as forcings for our dataset and the flood extent forecasting task, then these partitions ensure no data leakage across train and test splits. A map of

the basin partitions is illustrated in Fig. 2, with the five splits color-coded.

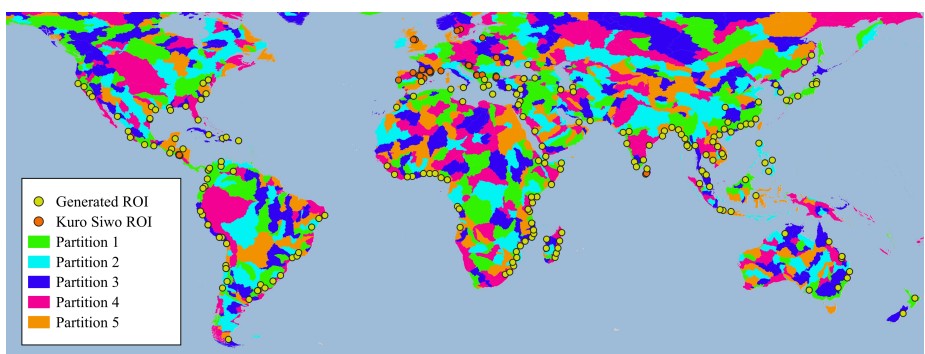

Figure 2: **Map** of hydrological basins. All level 4 HydroATLAS basins are split into five partitions (each represented by one of five colors) for the defined 5-fold cross-validation experimental protocol utilized in our benchmark.

**Are there any errors, sources of noise, or redundancies in the dataset?** If so, please provide a description.

The post-event mapping algorithm, although we build upon the state-of-the-art and apply further post-processing, is not flawless. While being more sophisticated than established heuristic approaches, our exchange with operational flood forecasting teams clarified that a separate and regularly re-running of permanent water detection is oftentimes employed in practice. Accordingly, we recommend a combination of our approach with such a complementary step to meet the best forecasting practices.

**Is the dataset self-contained, or does it link to or otherwise rely on external resources (e.g., websites, tweets, other datasets)?** If it links to or relies on external resources, a) are there guarantees that they will exist, and remain constant, over time; b) are there official archival versions of the complete dataset (i.e., including the external resources as they existed at the time the dataset was created); c) are there any restrictions (e.g., licenses, fees) associated with any of the external resources that might apply to a future user? Please provide descriptions of all external resources and any restrictions associated with them, as well as links or other access points, as appropriate.

The data is provided as a self-contained download hosted on Zenodo, available under https://zenodo.org/records/13133267. The original data sources have permissive licenses that make the data freely available for research purposes. This is not true for commercial purposes. The dataset should not be used for commercial purposes without the permission of the respective data owners.

- Sentinel-1: *'The access and use of Copernicus Sentinel Data and Service Information is regulated under EU law. In particular, the law provides that users shall have a free, full and open access to Copernicus Sentinel*

*Data and Service Information without any express or implied warranty, including as regards quality and suitability for any purpose'* https://sc ihub.copernicus.eu/twiki/do/view/SciHubWebPortal/TermsCondit ions (accessed 2024-05-03)

- CopDEM30: *'Copernicus data and information policy, regulated under European law2, ensures access on a full, open and free-of-charge basis as a rule with rare exceptions when needed to protect the security interest of the Union and its Member States as well as third party IPRs' [...] 'This licence concerns the use of the Copernicus WorldDEM-30 and for the reason stated above it is important to make available this Copernicus WorldDEM-30 to as many users as possible, Copernicus is therefore making it available on a free basis for the general public under the terms and conditions of this Licence. '* https://docs.sentinel-hub.com/api/latest/static/fil es/data/dem/resources/license/License-COPDEM-30.pdf (accessed 2024-05-29)

- Global 30m HAND: Creative Commons Zero (CC0) 1.0 Universal License, https://gis.asf.alaska.edu/arcgis/rest/services/GlobalHAND/G LO30_HAND/ImageServer (accessed 2024-05-03)

- ERA5 and ERA5 Land: *'4.1. This Licence is free of charge, worldwide, non-exclusive, royalty free and perpetual. '...' 4.2. Access to Copernicus Products is given for any purpose in so far as it is lawful, whereas use may include, but is not limited to: reproduction; distribution; communication to the public; adaptation, modification and combination with other data and information; or any combination of the foregoing.'* https://apps.e cmwf.int/datasets/licences/copernicus/ (accessed 2024-05-03)

- GloFAS: CEMS-FLOODS datasets licence *'[...] users are granted free access to the data of CEMS EFAS GloFAS for the following purposes and within the limits allowed under applicable law: (a) reproduction; (b) distribution; (c) communication to the public; (d) adaptation, modification and combination with other data and information; (e) any combination of points (a) to (d).'* https://cds.climate.copernicus.eu/api/v2/ter ms/static/cems-floods.pdf (accessed 2024-07-30)

- HydroATLAS: Creative Commons Attribution (CC-BY) 4.0 International License https://www.hydrosheds.org/hydroatlas (accessed 2024-05-03)

- Kuro Siwo: MIT License, https://github.com/Orion-AI-Lab/KuroSiw o?tab=MIT-1-ov-file (accessed 2024-05-03)

- Dartmouth Flood: Creative Commons Attribution (CC-BY) 4.0 International License https://floodobservatory.colorado.edu/WebMapServ erDataLinks.html (accessed 2024-05-03)

**Does the dataset contain data that might be considered confidential (e.g., data that is protected by legal privilege or by doctor-patient confidentiality, data that includes the content of individuals non-public communications)?** If so, please provide a description.

The dataset contains only freely available data and derivative works. Thus, there is no data that might be considered confidential.

**Does the dataset contain data that, if viewed directly, might be offensive, insulting, threatening, or might otherwise cause anxiety?** If so, please describe why.

There is no risk of accidentally viewing something distressing in this dataset. However, floods can cause significant devastation; if one was personally affected by such an event, they may be reminded of such events by viewing the floodmaps.

**Does the dataset relate to people?** If not, you may skip the remaining questions in this section.

This dataset does not relate to individuals or any subgroups, but it indirectly related to people as we identify natural disasters (as compared to non-hazardous, controlled inundation) by filtering the DFO records for counts of people harmed (i.e. displaced or killed).

**Does the dataset identify any subpopulations (e.g., by age, gender)?** If so, please describe how these subpopulations are identified and provide a description of their respective distributions within the dataset.

No subpopulations are identified. The identification properties we applied for our data are external to human-centric characteristics.

**Is it possible to identify individuals (i.e., one or more natural persons), either directly or indirectly (i.e., in combination with other data) from the dataset?** If so, please describe how.

It is not possible to identify individuals either directly or indirectly.

**Does the dataset contain data that might be considered sensitive in any way (e.g., data that reveals racial or ethnic origins, sexual orientations, religious beliefs, political opinions or union memberships, or locations; financial or health data; biometric or genetic data; forms of government identification, such as social security numbers; criminal history)?** If so, please provide a description.

The dataset does not contain any sensitive information.

**Any other comments?**

None.

**How was the data associated with each instance acquired?** Was the data directly observable (e.g., raw text, movie ratings), reported by subjects (e.g., survey responses), or indirectly inferred/derived from other data (e.g., part-of-speech tags, model-based guesses for age or language)? If data was reported by subjects or indirectly inferred/derived from other data, was the data validated/verified? If so, please describe how.

The input data associated with each instance was collected from large public archives. The target data associated with each instance (the floodmaps) was created by running a model trained on the Kuro Siwo (5) rapid flood mapping dataset and then post-processed to remove artifacts. The data from public archives is validated in the associated publications and externally, but not internally by us in terms of an additional processing step. The target data from Kuro Siwo was introspected and postprocessed to improve consistency over large areas.

**What mechanisms or procedures were used to collect the data (e.g., hardware apparatus or sensor, manual human curation, software program, software API)?** How were these mechanisms or procedures validated?

The data was collected using software and Application Programming Interface (API) calls. The generated floodmaps were manually curated to exclude failed processing.

**If the dataset is a sample from a larger set, what was the sampling strategy (e.g., deterministic, probabilistic with specific sampling probabilities)?**

The dataset is a subset of several large public archives. In particular, many sites and times were selected, and the respective data for each site and time was collected from those large public archives. The sites andtimes were chosen by intersecting the Dartmouth Flood Observatory (DFO) flood archive with HydroATLAS level 8 basins to choose potentially flooded locations. These basins were sorted by flood impact (weighted count of attributes $DISPLACED * 20000 + DEAD$) given by the DFO and basin population as reported by HydroATLAS in order to focus on natural disasters rather than non-hazardous inundation. For each flood $\times$ basin pair, Sentinel-1 images were searched. The pairs were then filtered by availability of Sentinel-1 images and to ensure that no two sites were too close in space and time. This continued until either there were no more flood $\times$ basin pairs left or the expected distribution of climate zones was met.

**Who was involved in the data collection process (e.g., students, crowdworkers, contractors) and how were they compensated (e.g., how much were crowdworkers paid)?**

Since the data collection process is automated, only the authors of the accompanying scientific publication were involved in the data collection process.

**Over what timeframe was the data collected? Does this timeframe match**

**the creation timeframe of the data associated with the instances (e.g., recent crawl of old news articles)?** If not, please describe the timeframe in which the data associated with the instances was created.

The dataset was collected from public archives over the months January 2024 to April 2024, and includes events from November 2014 (the launch of Sentinel-1) until August 2021 (the end of the flood archive).

**Were any ethical review processes conducted (e.g., by an institutional review board)?** If so, please provide a description of these review processes, including the outcomes, as well as a link or other access point to any supporting documentation.

No ethical review processes were conducted.

**Does the dataset relate to people?** If not, you may skip the remaining questions in this section.

This dataset does not relate to individual people.

**Did you collect the data from the individuals in question directly, or obtain it via third parties or other sources (e.g., websites)?**

Not applicable, see answers provided above.

**Were the individuals in question notified about the data collection?** If so, please describe (or show with screenshots or other information) how notice was provided, and provide a link or other access point to, or otherwise reproduce, the exact language of the notification itself.

Not applicable, see answers provided above.

**Did the individuals in question consent to the collection and use of their data?** If so, please describe (or show with screenshots or other information) how consent was requested and provided, and provide a link or other access point to, or otherwise reproduce, the exact language to which the individuals consented.

Not applicable, see answers provided above.

**If consent was obtained, were the consenting individuals provided with a mechanism to revoke their consent in the future or for certain uses?** If so, please provide a description, as well as a link or other access point to the mechanism (if appropriate).

Not applicable, see answers provided above.

**Has an analysis of the potential impact of the dataset and its use on data subjects (e.g., a data protection impact analysis) been conducted?** If so, please provide a description of this analysis, including the outcomes, as well as a link or other access point to any supporting documentation.

Not applicable, see answers provided above.

**Any other comments?**

Not applicable, see answers provided above.

**Was any preprocessing/cleaning/labeling of the data done (e.g., discretization or bucketing, tokenization, part-of-speech tagging, SIFT feature extraction, removal of instances, processing of missing values)?** If so, please provide a description. If not, you may skip the remainder of the questions in this section.

The primary value of this dataset is cessing existing data sources into a form easily usable by modern computer vision algorithms. Generally, the data from the public archives are collected, cessed, resampled and aligned with each site. More specifically:

- **Flood extent segmentation labels** are generated using models created with models trained on Kuro Siwo (5). There are four post-processing steps applied after generation to improve reliability and consistency:

  1. **Mosaicing artifact removal.** The models can only run on a $224 \times 224$ px tile at once. To circumvent mosaicing artifacts, the models are run on overlapping windows and the logits are smoothly transitioned between adjacent tiles. This technique removes all tiling artifacts.

  2. **Speckle denoising.** The resulting maps were processed with a majority vote filter (w = h = 5px) to remove speckled predictions

  3. **Contour filling.** Small holes were filled by running a contour-finding algorithm, then removing holes smaller than 50 px in area.

  4. **Permanent water label post-processing.** To improve the delineation between flood and permanent water pixels in the ensembled generated maps, we collide both classes into the first and superimpose the permanent water labels from ESA WorldCover (25) for the latter. This was found to improve separation of both water classes.

- The **Sentinel-1** images were processed following the procedure described in Kuro Siwo. Sentinel-1 GRD images are geolocated via an Orbit file, border and thermal are noise removed, calibrated, speckle filtered via a Lee Sigma filter ($\sigma = 0.9$, kernel size= 7) and terrain corrected onto SRTM 3 arcsecond DEM, in that order, using SNAP 9.0. The redistributed Sentinel-1 images in this dataset are additionally compressed with LERC and a maximum error of $10^{-4}$. This was validated by running the generating model on compressed and uncompressed Sentinel-1 images; there was an mIoU of around 99.7%.

- The **HydroATLAS** data used in this dataset is rasterised and provided as a tif file. Three types of variables were identified and handled differently for rasterisation:

  1. **Class variables** (e.g. climate zone) are rasterised by collecting fractional votes for each class variable within each pixel. Each basin's vote was scaled by the proportion of the pixel that the basin covered.

2. **Absolute/summed variables** (e.g. yearly streamflow) are rasterised by summing up the contribution from each basin within each pixel. The basin's contribution is the proportion of the basin covered by the pixel multiplied by the basin variable value.

3. **Relative/averaged variables** (e.g. average temperature) are rasterised by calculating a weighted sum of the values in the basins which overlapped the pixel; weighted by the proportion of the pixel the basin covered.

- The **ERA5 and ERA5-Land** rasters are daily composites, downloaded as-is from Google Earth Engine. Likewise, the **GloFAS** rasters where downloaded as daily aggregates from the Copernicus Climate Data Store and subsequently sliced into patches matching the temporal extent and spatial coverage of ERA5 and other context-level data.

**Was the "raw" data saved in addition to the cessed/cleaned/labeled data (e.g., to support unanticipated future uses)?** If so, please provide a link or other access point to the "raw" data.

The raw data is all freely available. If intermediate values are needed again, they can be regenerated with our published scripts. However, this may take a significant amount of processing time.

**Is the software used to preprocess/clean/label the instances available?** If so, please provide a link or other access point.

The code used to create the dataset is freely available at `https://github.com/Multihuntr/GFF`. Beyond providing all software needed to reproduce the dataset curation, it can also be utilized to extend the existing dataset with additional information and data modalities as well as novel ROI or events.

**Any other comments?**

None.

**Has the dataset been used for any tasks already?** If so, please provide a description.

The dataset has not been utilized prior to its design and creation for the associated scientific publication.

**Is there a repository that links to any or all papers or systems that use the dataset?** If so, please provide a link or other access point.

The dataset repository contains training scripts to train models on the dataset. The dataset repository is: https://github.com/Multihuntr/GFF. Should the dataset be adopted by future end users which would like to be referenced, then we may list the adopters of this dataset in a separate section of the repository. This should be indicated by interested users making a pull request to update the list.

**What (other) tasks could the dataset be used for?**

The dataset implies modelling a single step from weather to flood extent. With the inclusion and the optional usage of historical GloFAS data, the user can aim for a two-step approach where the task then is to map the pre-modelled streamflow and river-runoff inputs to the desired flood extent forecasts. In the future, others may choose to predict streamflow using their own data, and then from streamflow predict flood extent using this dataset.

**Is there anything about the composition of the dataset or the way it was collected and cessed/cleaned/labeled that might impact future uses?** For example, is there anything that a future user might need to know to avoid uses that could result in unfair treatment of individuals or groups (e.g., stereotyping, quality of service issues) or other undesirable harms (e.g., financial harms, legal risks) If so, please provide a description. Is there anything a future user could do to mitigate these undesirable harms?

The flood segmentation annotations, even though involving several steps of post-processing, are generated in an automated manner via machine learning algorithms. As stated in the accompanying scientific publication's main text, *'the models classify no-water pixels with F1 scores of $99.07$ and $98.97$ and water pixels with F1 scores of $87.58$ and $86.52$ on hold-out data, respectively'* [5]. However, it should be critically evaluated whether this level of label goodness matches the quality of expert annotations and the needs of safety-critical settings. In general, good performance on this dataset may not guarantee good performance in general, even though our experimental setup follows best practices to assess generalization. Therefore, further studies on the particular regions of interest should be conducted before deploying models trained on this dataset in practice.

**Are there tasks for which the dataset should not be used?** If so, please provide a description.

See above.

**Any other comments?**

None.

**Will the dataset be distributed to third parties outside of the entity (e.g., company, institution, organization) on behalf of which the dataset was created?** If so, please provide a description.

The dataset will be freely available for research purposes, worldwide.

**How will the dataset will be distributed (e.g., tarball on website, API, GitHub)** Does the dataset have a digital object identifier (DOI)?

The dataset will be provided through Zenodo and its release will be linked to a DOI. The dataset is available under https://zenodo.org/records/13133267.

**When will the dataset be distributed?**

The dataset is available via https://zenodo.org/records/13133267 and will be released as a final version upon acceptance of the accompanying submitted scientific paper.

**Will the dataset be distributed under a copyright or other intellectual property (IP) license, and/or under applicable terms of use (ToU)?** If so, please describe this license and/or ToU, and provide a link or other access point to, or otherwise reproduce, any relevant licensing terms or ToU, as well as any fees associated with these restrictions.

All of our code, models and derived data such as the generated floodmaps will be provided with a CC0 license. However, the public archive data sources which we sourced have their own licenses, which are very permissive but need to be respected.

**Have any third parties imposed IP-based or other restrictions on the data associated with the instances?** If so, please describe these restrictions, and provide a link or other access point to, or otherwise reproduce, any relevant licensing terms, as well as any fees associated with these restrictions.

The public archives used to create input data have their own licences. They all allow free use for research purposes. Some do not allow commercial uses by default:

- Sentinel-1: *'The access and use of Copernicus Sentinel Data and Service Information is regulated under EU law. In particular, the law provides that users shall have a free, full and open access to Copernicus Sentinel Data and Service Information without any express or implied warranty, including as regards quality and suitability for any purpose'* https://scihub.copernicus.eu/twiki/do/view/SciHubWebPortal/TermsConditions (accessed 2024-05-03)

- CopDEM30: *'Copernicus data and information policy, regulated under European law2, ensures access on a full, open and free-of-charge basis as a rule with rare exceptions when needed to protect the security interest of the Union and its Member States as well as third party IPRs' [...] 'This licence concerns the use of the Copernicus WorldDEM-30 and for the reason*

*stated above it is important to make available this Copernicus WorldDEM-30 to as many users as possible, Copernicus is therefore making it available on a free basis for the general public under the terms and conditions of this Licence. '* https://docs.sentinel-hub.com/api/latest/static/files/data/dem/resources/license/License-COPDEM-30.pdf (accessed 2024-05-29)

- Global 30m HAND: Creative Commons Zero (CC0) 1.0 Universal License, https://gis.asf.alaska.edu/arcgis/rest/services/GlobalHAND/GLO30_HAND/ImageServer (accessed 2024-05-03)

- ERA5 and ERA5 Land: *'4.1. This Licence is free of charge, worldwide, non-exclusive, royalty free and perpetual. '...' 4.2. Access to Copernicus Products is given for any purpose in so far as it is lawful, whereas use may include, but is not limited to: reproduction; distribution; communication to the public; adaptation, modification and combination with other data and information; or any combination of the foregoing.'* https://apps.ecmwf.int/datasets/licences/copernicus/ (accessed 2024-05-03)

- GloFAS: CEMS-FLOODS datasets licence *'[...] users are granted free access to the data of CEMS EFAS GloFAS for the following purposes and within the limits allowed under applicable law: (a) reproduction; (b) distribution; (c) communication to the public; (d) adaptation, modification and combination with other data and information; (e) any combination of points (a) to (d).'* https://cds.climate.copernicus.eu/api/v2/terms/static/cems-floods.pdf (accessed 2024-07-30)

- HydroATLAS: Creative Commons Attribution (CC-BY) 4.0 International License https://www.hydrosheds.org/hydroatlas (accessed 2024-05-03)

- Kuro Siwo: MIT License https://github.com/Orion-AI-Lab/KuroSiwo?tab=MIT-1-ov-file (accessed 2024-05-03)

- Dartmouth Flood: Creative Commons Attribution (CC-BY) 4.0 International License https://floodobservatory.colorado.edu/WebMapServerDataLinks.html (accessed 2024-05-03)

**Do any export controls or other regulatory restrictions apply to the dataset or to individual instances?** If so, please describe these restrictions, and provide a link or other access point to, or otherwise reproduce, any supporting documentation.

No export controls or regulatory restrictions apply to this dataset and, to the authors' best knowledge, neither to any of the precursor products this dataset builds upon.

**Any other comments?**

None.

**Who will be supporting/hosting/maintaining the dataset?**
The dataset is hosted on Zenodo, which offers DOI for datasets and long-term availability. The code to generate and extend the dataset as well as to reproduce the benchmarks is hosted on GitHub. Both hosting solutions are established platforms and are expected to provide long-term maintenance. The authors will be providing support with the dataset through GitHub, e.g. resolving pull requests and open issues. Furthermore, a leaderboard of the best (peer-reviewed) models will be maintained in the repository's README.

**How can the owner/curator/manager of the dataset be contacted (e.g., email address)?**
If there is an issue with the dataset, they can be raised as an issue on the GitHub repository.

**Is there an erratum?** If so, please provide a link or other access point.
There are no errata (yet).

**Will the dataset be updated (e.g., to correct labeling errors, add new instances, delete instances)?** If so, please describe how often, by whom, and how updates will be communicated to users (e.g., mailing list, GitHub)?
Both the Zenodo and GitHub repositories will be updated with any changes as they are addressed. There is no fixed schedule to address these, nor is there time pre-allocated to fixing such issues. As such, updates may be focused to address the most critical issues to ensure reproducibility.

**If the dataset relates to people, are there applicable limits on the retention of the data associated with the instances (e.g., were individuals in question told that their data would be retained for a fixed period of time and then deleted)?** If so, please describe these limits and explain how they will be enforced.
The dataset does not relate to individual people.

**Will older versions of the dataset continue to be supported/hosted / maintained?** If so, please describe how. If not, please describe how its obsolescence will be communicated to users.
Zenodo and GitHub are automatically versioned. Zenodo enforces a versioning scheme, allowing old versions to be downloaded and showing the latest version.

**If others want to extend/augment/build on/contribute to the dataset, is there a mechanism for them to do so?** If so, please provide a description. Will these contributions be validated/verified? If so, please describe how. If not, why not? Is there a process for communicating/distributing these contributions to other users? If so, please provide a description.
Such a motivated person could use our code base to generate more floodmaps at new locations. If such a change significantly improves the dataset, they could

also submit a GitHub pull request. The request would be reviewed and manually verified by the authors.

### Any other comments?

For further questions or remarks, kindly reach out to the authors of this datasheet and the associated scientific work.

# 3  Supplementary Material

## 3.1  Extended metrics for baseline benchmarking

To complement the baseline benchmarks in the main text with additional analysis, the F1 scores are complemented with metrics of Intersection over Union (IoU) and Accuracy. All metrics are provided as overall scores, as well as specifically evaluated on the background (B) and water (W) classes.

Table 1: **Extended metrics** for all five baselines, evaluated on every ROI. Metrics tend to agree besides minor differences, with U-TAE performing best.

| Model | F1 | F1-B | F1-W | IoU | IoU-B | IoU-W | Accuracy | Accuracy-B | Accuracy-W |
|---|---|---|---|---|---|---|---|---|---|
| U-TAE (7) | **0.77 ± 0.04** | **0.97 ± 0.00** | **0.57 ± 0.07** | **0.67 ± 0.04** | **0.94 ± 0.01** | **0.40 ± 0.07** | **0.81 ± 0.06** | **0.96 ± 0.01** | 0.65 ± 0.14 |
| LSTM U-Net (19) | 0.76 ± 0.04 | **0.97 ± 0.00** | 0.55 ± 0.08 | 0.66 ± 0.04 | **0.94 ± 0.01** | 0.39 ± 0.07 | 0.80 ± 0.07 | **0.96 ± 0.01** | 0.65 ± 0.14 |
| 3DConv U-Net (19) | 0.76 ± 0.04 | 0.97 ± 0.01 | 0.54 ± 0.08 | 0.66 ± 0.04 | **0.94 ± 0.01** | 0.38 ± 0.07 | 0.79 ± 0.07 | 0.96 ± 0.02 | 0.63 ± 0.16 |
| MaxViT U-Net (24; 3) | 0.75 ± 0.03 | 0.96 ± 0.01 | 0.53 ± 0.06 | 0.65 ± 0.04 | 0.93 ± 0.02 | 0.37 ± 0.06 | 0.80 ± 0.05 | 0.95 ± 0.03 | 0.65 ± 0.12 |
| logistic regression (13) | 0.66 ± 0.04 | 0.93 ± 0.02 | 0.40 ± 0.07 | 0.56 ± 0.04 | 0.87 ± 0.02 | 0.26 ± 0.06 | 0.80 ± 0.06 | 0.88 ± 0.03 | **0.72 ± 0.13** |

## 3.2  Interpretation of regression feature weights

Following the approach of (13; 9), we implement a logistic regression model to provide a simple analysis of feature importance. Analogous to the prior work of (9), we implement the method as a single-layer convolutional network with $3 \times 3$ kernels and train it on the mono-temporal local information. The weights are accumulated by computing the kernel means and reported in Table 2.

Positive values drive classification of pixels as background, whereas negative values contribute to water classification. The simple regression model assigns greatest importance to the S1 VV channel, which is a major driver of background classification. Notably, only the S1 VH class contributes to water classification. Finally, it is noteworthy that the model considered HAND more important than DEM, even in the absence of any context-scale precipitation forecasts.

| Modality | Weight |
|---|---|
| S1 VV | 2.47 |
| S1 VH | -0.96 |
| DEM | 0.04 |
| HAND | 0.13 |

Table 2: **Importance analysis** of local scale information, as learned by the simple logistic regression model. Positive values contribute to classification of background, while negative weights drive water prediction.

## 3.3 Ablation Experiments

To explore the role of expert-annotated as well as generated target labels and how they drive model performance, the following experiment conducts ablations. Specifically, U-TAE is re-trained in three conditions: only on ROI for which Kuro Siwo labels are available, only on generated labels for all other ROI, and finally on all ROI including all annotation types. The outcomes are presented in Table 3. The results show that training on generated labels generally improves the performance. While this indicates that the generated labels are of sufficient quality, one should also consider that they constitute a substantially larger quantity of data. Overall, the experiment implies that usage of the generated labels is beneficial on generated as well as hand-annotated data.

Table 3: **Ablation studies** by restricting training examples for the best-performing model, U-TAE (7). The full dataset includes both Kuro Siwo hand-labelled ROIs and generated ROIs. This outcome underlines the value of using both.

| Model | F1 | F1-B | F1-W | $F1_{KS}$ | F1-$B_{KS}$ | F1-$W_{KS}$ |
|---|---|---|---|---|---|---|
| Kuro Siwo labels | $0.64 \pm 0.03$ | $0.94 \pm 0.02$ | $0.35 \pm 0.06$ | $0.66 \pm 0.08$ | $0.93 \pm 0.09$ | $0.39 \pm 0.07$ |
| Generated labels | $0.75 \pm 0.02$ | $0.96 \pm 0.00$ | $0.53 \pm 0.05$ | $0.69 \pm 0.08$ | $0.97 \pm 0.01$ | $0.42 \pm 0.14$ |
| Full dataset | $\mathbf{0.77 \pm 0.04}$ | $\mathbf{0.97 \pm 0.00}$ | $\mathbf{0.57 \pm 0.07}$ | $\mathbf{0.72 \pm 0.02}$ | $\mathbf{0.97 \pm 0.01}$ | $\mathbf{0.48 \pm 0.05}$ |

An ablation study on the input modalities is conducted by systematically knocking out individual forcings of the best-performing model of table 1. This is to explore the importance of every product included in our dataset. The outcomes are presented in Table 4 and highlight the benefit of each curated input modality. Due to the complexity of the task and the multi-causal nature of the floods featured in the curated data, each observation type is beneficial.

Table 4: **Ablation studies** by systematically knocking out individual input features for the best-performing model, U-TAE (7). The outcomes underline the value of each modality, with the absence of S1 having the biggest impact.

| Model | F1 | F1-B | F1-W | $F1_{KS}$ | F1-$B_{KS}$ | F1-$W_{KS}$ |
|---|---|---|---|---|---|---|
| all modalities | $\mathbf{0.77 \pm 0.04}$ | $\mathbf{0.97 \pm 0.00}$ | $\mathbf{0.57 \pm 0.07}$ | $0.72 \pm 0.02$ | $\mathbf{0.97 \pm 0.01}$ | $0.48 \pm 0.05$ |
| no ERA5 & ERA5-Land | $0.76 \pm 0.04$ | $0.97 \pm 0.01$ | $0.55 \pm 0.08$ | $0.71 \pm 0.12$ | $\mathbf{0.97 \pm 0.01}$ | $0.45 \pm 0.22$ |
| no HydroATLAS | $0.76 \pm 0.04$ | $\mathbf{0.97 \pm 0.00}$ | $0.55 \pm 0.07$ | $0.71 \pm 0.03$ | $0.96 \pm 0.03$ | $0.46 \pm 0.04$ |
| no HAND | $0.76 \pm 0.06$ | $0.96 \pm 0.01$ | $0.56 \pm 0.12$ | $0.72 \pm 0.05$ | $\mathbf{0.97 \pm 0.01}$ | $0.47 \pm 0.11$ |
| no DEM | $0.75 \pm 0.04$ | $0.96 \pm 0.01$ | $0.54 \pm 0.08$ | $\mathbf{0.73 \pm 0.05}$ | $0.97 \pm 0.02$ | $\mathbf{0.50 \pm 0.10}$ |
| no S1 | $0.70 \pm 0.08$ | $0.95 \pm 0.02$ | $0.44 \pm 0.17$ | $0.69 \pm 0.09$ | $\mathbf{0.97 \pm 0.01}$ | $0.41 \pm 0.19$ |