# OpenReview forum: "Off to new Shores: A Dataset & Benchmark for (near-)coastal Flood Inundation Forecasting"
_NeurIPS.cc/2024/Datasets_and_Benchmarks_Track — NeurIPS 2024 Track Datasets and Benchmarks Poster_

### Official Review · Reviewer_vRcU · 2024-07-25
**A dataset for flood extent mapping and forecasting**

**Rating:** 9
**Confidence:** 4
**Correctness:** All claims sound correct and the data…

**Review:**

This work is technically sound and of high quality. The authors address the challenging task---and for which there are less datasets available--of flood extent forecasting. The pros are listed in the strengths section below, and the cons are listed in the opportunities for improvements and limitations/clarity.

**Strengths:**

The strengths of this work are: 1) the integration of multi-temporal and multi-modal data (atmospheric reanalysis products, high resolution terrain models and simulated precipitation drainage, hydrological basin attributes, pre-flood Sentinel-1); 2) global representativeness of coastal areas prone to flooding and their varying climate zones; and 3) the contribution to separating coastal floods and  near-coastal/inland floods.

**Additional Feedback:**

None

**Clarity:**

The paper is very well written. Below, I list a few minor comments on possible improvements on clarity:

Fig 1:
	- the red dots in Column 1–3 are hard to see
	- the caption should mention what are the red and blue regions in Column 6.

L99: Floodmaps were not generated for a ROI within < 5◦ proximity of a previously selected ROI or if its climate zone is already represented sufficiently
	- it is unclear why an ROI within 5◦ might be excluded if the flood event occurred at different times

L120: while images during the event are used for the label generation.
	- It would be useful to specify how the event duration (end/beginning) was sourced and/or determined---likely from DFO but that could be made explicit

Section 3.1: ROIs
	- it would useful to specify how many of the included 298 events are originally from the 280 in the tropical cyclones dataset, as that might indicate there is a bias towards flood events driven by storm surges, as opposed to driven by fluvial or pluvial.

L138: on hold-out data
	- specify how the hold-out data were selected; specifically, was data leakage avoided?

L214: with tiles categorized whether they are less or more than 30 km distant from the nearest coast.
	- any reference to support the 30 km threshold?

I think it would also be useful to mention if any steps were taken to address class imbalance (water vs non-water) and what was the loss function utilized in the benchmarks. These could be included in the supplemental if needed.

**Documentation:**

The main document and the supplemental material include a high level of detail to insure reproducibility. Links are provided to the code base, input and output datasets, including their licenses.

**Ethics:**

No concerns.

**Limitations:**

I see no potential negative societal impact. Other than that, the authors have adequately acknowledged the limitations or this work.

**Opportunities For Improvement:**

The authors already recognize and list the study limitations (section 5), I don't have any other to add. My only minor comment for something that could be added to discussion:

L247: there is a trend of performance decrease
	- hypothesize why?

**Relation To Prior Work:**

Yes, this work is well contextualized.

**Summary And Contributions:**

This work presents a dataset primed for both rapid flood extent mapping as well as flood extent forecasting. The data included are based on Sentinel-1 observations, along with other modalities which are used for the forecasting task: weather, topography and hydrography. The authors also include a two-parts benchmark, one for global floods mapping and another one specific for coastal flood events.

---

> ### Author Rebuttal · Authors · 2024-08-18
>
> ## Dear reviewer vRcU,
>
> Thanks once more for your constructive feedback, your peer review and suggestions helped us improve on our initial submission. In the following, please find our rebuttal to your review and please also have a look at the main post outlining all further improvements. We hope our answers to your questions and suggestions confirm your positive sentiment.
>
> > L247: there is a trend of performance decrease - hypothesize why?
>
> It’s instructive to keep in mind that the Kuro Siwo ROI constitute less than 15 % of the full dataset. They pertain to ROI that are inland, more distant from coastlines --- in comparison to the other 85 % of the dataset. That is, the Kuro Siwo ROI may display different characteristics, such as soil properties (as characterized by the HydroATLAS layers) and climate (as provided by ERA5 and ERA5-Land). Altogether, these differences may translate into different flood event dynamics and we hypothesize that this may explain performance differences.
>
> > Fig 1: - the red dots in Column 1–3 are hard to see - the caption should mention what are the red and blue regions in Column 6.
>
> We revised the figure by including a legend at the figure’s right side to clarify the color-coding. Furthermore, with the re-sampling of context information from 0.1 to 0.05 degrees, the context information is center-cropped (to preserve prior shapes and dimensions) and the red dots are now more visible. We hope this sufficiently improves on the figure’s legibility.
>
>
> > L99: Floodmaps were not generated for a ROI within < 5◦ proximity of a previously selected ROI or if its climate zone is already represented sufficiently - it is unclear why an ROI within 5◦ might be excluded if the flood event occurred at different times
>
>
> For particular complex flood scenarios (e.g. compound events, or subsequent events), DFO can contain different entries at neighbouring and overlapping time points. Hence, there would be a potential for data leakage at the context level without further measures of cautiousness. We decided that it would be best to avoid any possible data leakage at all and therefore avoid ROI  too close to one another in either time or space.
>
>
> >  L120: while images during the event are used for the label generation. - It would be useful to specify how the event duration (end/beginning) was sourced and/or determined---likely from DFO but that could be made explicit
>
> Yes, that is correct. The coarse event duration is specified by the end and beginning intervals reported in the DFO catalogue. This interval is brief at times, but can be elongated depending on the nature and complexity of the flood event. At the longest, floods may last between a week and a month. Such durations then allow for repeated S1 revisits and in these cases there are two to three opportunities to observe flooding. Following the suggestions oft he review, we included the following information in the main text: *‘Flooded ROI may vary in their extent if DFO indicates long time intervals. In this case, multiple S1 passes may be available and we pick the timestamp exhibiting the largest flood extent.‘*
>
> > Section 3.1: ROIs - it would useful to specify how many of the included 298 events are originally from the 280 in the tropical cyclones dataset, as that might indicate there is a bias towards flood events driven by storm surges, as opposed to driven by fluvial or pluvial.
>
> 99 of the dataset’s 298 ROIs overlap with floods driven by tropical cyclones, reported in both DFO and [TH21]. We suppose that a small minority of these events is associated with storm surges, as the focus of [TH21] is on fluvial and pluvial floods and the corresponding DFO entries, albeit very coarse in detail, don’t list storm surges as a direct cause of floods either. We specify the numbers by including the following in the main text: *‘The outlined procedure results in 298 ROI of flood events depicted in the map of Fig. 3 (99 of which are listed by both DFO and the database of (68), indicating cyclone-driven floods).‘*
>
> > L138: on hold-out data - specify how the hold-out data were selected; specifically, was data leakage avoided?
>
>
> Data leakage is avoided by taking cautious measures at multiple stages:
>
> - **ROI selection**. When ROI candidates are incrementally selected for inclusion in the data set, no ROI closer than circa 500 km to one another are considered. That is, a ROI is only included in the dataset if it is at least 500 km apart from the closest neighbouring ROI. This is to exclude any overlap of contextual information, as given by e.g. ERA5 & ERA5-Land.
>
> - **Experimental Design**. Our overall experimental setup is a 5-fold cross-validation scheme. For each of the five separate runs, we use one partition as the test set, another as the validation set and the remaining three for training. The partitions are mutually exclusive in terms of their spatial coverage, i.e. there is no data leakage.
>
> [continuing in adjacent post]

---

> > ### Author Rebuttal · Authors · 2024-08-18
> >
> > [continuing in adjacent post]
> >
> >
> > > L214: with tiles categorized whether they are less or more than 30 km distant from the nearest coast. - any reference to support the 30 km threshold?
> >
> > Our apologies for the confusion, there was a slight inconsistency in the main text which we resolved now by correcting the cited sentence to: *‘with tiles categorized whether they are less or more than 10 km distant from the nearest coast’*
> >
> > This was a design choice, but it is well-motivated. The reason is that we wanted **both categories to be distinguished by at least a single grid cell of atmosphere information, as given by ERA5-Land.** ERA5-Land is given as time series data at the context scale with a horizontal resolution of 0.1°, which equals **roughly 10 km** at the mid-latitudes. We also considered to utilize ERA5’s 30 km at some point, but a priori decided that this would be too generous of a margin for a definition of areas meant to be strictly coastal. This is motivated in the main text as *‘Regions 10 km or closer to the nearest shore are considered as coastal, a distance about the size of one ERA5-Land cell.‘*
> >
> >
> > > I think it would also be useful to mention if any steps were taken to address class imbalance (water vs non-water) and what was the loss function utilized in the benchmarks. These could be included in the supplemental if needed.
> >
> > The cost function is a cross entropy loss with class weightings of 0.5 and 5 (could be rescaled in interaction with the learning rate, this is just for historical reasons). We appreciate the thoughtful suggestion and inserted the following information in the main text: *‘The cost function is a binary cross entropy loss with class weightings of 0.5 and 5 for background versus water classes --- roughly equal to each category's inverse frequency. Using the same loss, models are evaluated on the validation partition every epoch […]’*

---

> > > ### Comment · Reviewer_vRcU · 2024-08-31
> > > **Superb rebuttals, wanted to increase score +1**
> > >
> > > I am very satisfied with the rebuttals and clarifications the authors have added, with high level of detail, to my comments as well as to the other reviewers comments. Therefore, I wanted to update my scores to 9. Unfortunately, it seems neurips closed the option for editing reviews before the published deadline—at least since yesterday (2 days before the deadline) I could no longer see the “edit” button.

---

### Official Review · Reviewer_ZKym · 2024-07-27
**Well-designed dataset for important task, but concerns about pseudolabels**

**Rating:** 5
**Confidence:** 5

**Review:**

Pros:
- The GFF dataset and accompanying benchmark address an important task of flood detection and forecasting.
- The dataset and benchmark tasks are thoughtfully designed (see Strengths).

Cons:
- The dataset labels combine expert annotations from Kuro Siwo and pseudolabels. The accuracy of the pseudolabels seems like it would place an upper limit on the accuracy that can be achieved by models trained and tested on the dataset.

**Strengths:**

The dataset and benchmark tasks are thoughtfully designed. I appreciated several attributes such as:
- It seems that the dataset is geographically balanced with the same number of events at each ROI, so performance metrics are not biased toward one particular ROI
- The authors took care to avoid spatial or temporal autocorrelation in the splits
- The cross validation setup using HydroATLAS basins is well thought out
- The F1 score is only computed for pixels that are not permanent water bodies, which makes the metric more sensitive to the actual flood water pixels that are targeted
- The authors separate the performance on the KS labels (expert annotation) from the performance on all labels (not clear if this is part of the benchmark, but IMHO it should be)

**Additional Feedback:**

- In Table 1, W is used for water pixels on the left and F for flood pixels on the right. Are these meant to be the same?
- The paper states, "To our knowledge, there are no existing solutions processing such significant differences in scale" referring to ERA5 and S1. Presto includes ERA5 and S1: https://arxiv.org/abs/2304.14065

**Clarity:**

The paper is well written and easy to read.

It seems the purpose of Figure 5 is to justify that the pseudolabels (and expert annotations) generally match the distribution of floods expected from DFO. Differences from DFO are attributed to the near-coastal focus of this dataset. How else could differences arise, e.g., quality of pseudolabels?

**Correctness:**

The claims are correct and the dataset is structured in a sound way. Experiments seems to have been performed correctly.

**Documentation:**

I did not see a maintenance plan for the dataset

**Ethics:**

No ethics concerns

**Limitations:**

The main limitation of this dataset is that it treats pseudolabels as ground truth labels.
- It is not clear which parts, or how much, of the dataset comes from pseudolabels vs expert annotation (see questions/suggestions above).
- The pseudolabels are generated from a model trained on the expert KS labels. The accuracy reported for water pixels is 87.58 on training samples and 86.52 on held-out samples. This seems like a fairly low upper limit to place on performance models can expect to achieve on the benchmark. In addition, we do not have sufficient information to assess our confidence in those metrics. How were the KS labels split in that experiment? Are the regions in the heldout samples similar to those in the training samples? Are the heldout samples in regions similar to those in the pseudolabeled ROIs or is there a significant distribution shift?

**Opportunities For Improvement:**

The following are ways that I think this paper and proposed dataset/benchmark can be brought above the acceptance threshold to address my concerns about the pseudolabels:
- In Figure 3, can you indicate which ROIs have KS labels vs pseudolabels? Can you also show the bounds of the HydroATLAS basins used for cross validation on this map?
- Can you review and correct all pseudolabeled masks so we can be confident that the masks are high quality? If not, how else can you convince readers that these masks are high quality? (The F1 scores reported for the water pixels give concerns about this)

In addition, it seems that there is high variability between the cross validation folds in Table 1-2. If I understand correctly, this suggests some models are performing much better in some regions (basins) than in others. It is hard to account for this range when comparing models. How can you modify the score across all folds to account for this (or should you), e.g., basing a model's score on the lowest fold?

**Relation To Prior Work:**

There are several flood detection datasets (e.g. Sen1Floods11, Sen12-FLOOD, etc). It would be helpful to have a more granular comparison of the differences between the GFF dataset and existing datasets (e.g., geographic coverage, number of samples, etc.)

**Summary And Contributions:**

The paper presents the GFF (Global Flood Forecasting) dataset including two benchmark tasks: flood detection and flood forecasting. While there are several existing flood detention benchmark datasets, this is the first benchmark for flood forecasting. The dataset covers global coastal/near-coastal regions and includes flood mask labels paired with multiple geospatial data layers: Sentinel-1, ERA5/ERA5-Land, DEM/HAND, and HydroATLAS basin attributes. Flood mask labels come from expert annotations in the Kuro Siwo (KS) dataset, augmented with pseudolabels from a model trained on the KS dataset. Experiments were performed to evaluate current state of the art with representative models including U-net based models for both the detection and forecasting tasks. Results showed performance gaps that can be addressed in future research using this benchmark and dataset.

---

> ### Author Rebuttal · Authors · 2024-08-18
>
> ## Dear reviewer ZKym,
>
> Thanks once more for your constructive feedback, your peer review and suggestions helped us improve on our initial submission. In the following, please find our rebuttal to your review and please also have a look at the main post outlining all further improvements. We hope our replies to your questions may clear any remaining concerns.
>
>
> > The dataset labels combine expert annotations from Kuro Siwo and pseudolabels. The accuracy of the pseudolabels seems like it would place an upper limit on the accuracy that can be achieved by models trained and tested on the dataset.
>
>
> We agree with the reviewer that the usage of model-generated labels **may induce an upper bound of performance on the considered task.** However, we’d like to raise the **following points in defence of the current approach:**
> - Even for hand-annotated labels there may be **inter-rater variability and no perfect agreement.** This is because of the tedious nature of the task, its complexity in spotting & deciding which pixels are flooded and the diversity of contractors and service providers typically involved (see e.g. the contractors for Copernicus EMS here https://emergency.copernicus.eu/mapping/list-of-activations-rapid). Citing [i], “different teams within Copernicus annotate various flood events, resulting in variable photointerpretation and methodological biases in the annotations“. We think that exploring inter-rater reliability is valuable to explore in this context and expect that ensembles of automated rapid mapping methods may prove valuable in this context.
> - The **validated accuracy of the generated labels is relatively high** (“the models classify no-water pixels with F1 scores of 99.07 and 98.97 and water pixels with F1 scores of 87.58 and 86.52 on hold-out data”, p.5 l. 137-138). A particular concern expressed in this review is on the 87.58 and 86.52 F1 scores on the water class, but this leaves plenty room for improvements on the currently accomplished baseline model scores of 0.43 - 0.54 on the same class. Accomplishing such performance on a rapid mapping task is a solid job, but obtaining similar scores in a forecasting task will be even more challenging. In this sense, we believe that the generated labels will offer a **meaningful runway for further model improvements.**
> - *“The accuracy reported for water pixels is 87.58 on training samples and 86.52 on held-out samples.”*
>   - Sorry, this is a misunderstanding which we may have caused but would wish to clarify. Both numbers of 87.58 and 86.52 pertain to hold-out sample F1 scores on the water pixel class. Please note of the following three points:
>     - We reference **two performance scores of two different models** here. We utilised both the best and the second-best pre-trained Kuro Siwo model to generate labels for every sample, and then formed an ensemble prediction by averaging the logits as reported in the first paragraph of section “3.3 Generating floodmaps anywhere”.
>     - Both scores pertain to **performance on hold-out samples**. There’s no train score reported here and we are interested in how well the pre-trained model generalises to unseen regions, such as the other ROI we consider.
>     - Both scores are given in terms of **F1, rather than accuracy.** This is worth considering, as accuracy doesn’t take false alarms into account. On the minority class (‘water pixels’) this potentially makes the F1 score the more strict metric of the two.
>
> In sum, we acknowledge the criticism wrt potential upper performance bounds, but wish to clarify that this bound offers much room for improving current forecasting approaches. For the future, we expect further improvements in rapid mapping and significant performance gains may motivate a follow-up to our current work.
>
>
> > Can you review and correct all pseudolabeled masks so we can be confident that the masks are high quality? If not, how else can you convince readers that these masks are high quality? (The F1 scores reported for the water pixels give concerns about this)
>
> Reviewing and editing the labels generated for all 298 ROI poses practical challenges. All generated masks were reviewed. But, unfortunately, none of the authors are formally trained as rapid mapping experts to interpret SAR data for flooding. We reached out to a few such experts, but none responded. Regardless, our review involved checking for plausibility (e.g. following ridges and known rivers), frequency of segmentation artefacts (e.g. gridding, sharp perfectly straight lines) and best guesses using Kuro Siwo labels as a guide. It was these observations that informed our post-processing, as specified in section “3.4 Label post-processing” to improve generated label quality.
> To further convince the readers that the generated labels are high quality we have undertaken a new experiment: We re-trained the best model of Table ‘Track 1’ using *generated labels only*, and *hand-annotated Kuro Siwo labels only*, and compare these to using the full dataset on the same test set. On the held-out Kuro Siwo test set, these models got a mean F1 of 0.42 and 0.39, respectively, across folds. Compared to the full dataset model which got 0.48. This highlights the value of using both a larger collection of generated labels and the hand-labelled masks from Kuro Siwo at the same time. We have added a full table of these results to the Supplementary Materials (submitted with the datasheet).
>
> We note that while working on the project we presented various sample generated maps to colleagues (not authors) when discussing what we were working on, as well as external visitors. Unfortunately none of them were formally trained in rapid mapping of SAR data either. But, they generally had the same opinion as us; the generated labels appeared plausible, and there were no critical artefacts.
>
> [continuing in adjacent post]

---

> > ### Author Rebuttal · Authors · 2024-08-18
> >
> > [continuing in adjacent post]
> >
> > > In addition, it seems that there is high variability between the cross validation folds in Table 1-2. If I understand correctly, this suggests some models are performing much better in some regions (basins) than in others. It is hard to account for this range when comparing models. How can you modify the score across all folds to account for this (or should you), e.g., basing a model's score on the lowest fold?
> >
> >
> > We agree that the baseline models display meaningful cross-split variability, but it is important to note that **variability seems inversely related to model performances**. That is, models being more performant on average also show a trend of greater consistency and less variance. This is noteworthy, as the **trend may imply further consistency increase with mean performance gains.** We’d like to see this as an outcome of future models benchmarked on GFF.
> >
> >
> > > It is not clear which parts, or how much, of the dataset comes from pseudolabels vs expert annotation (see questions/suggestions above). […] How were the KS labels split in that experiment? Are the regions in the heldout samples similar to those in the training samples? Are the heldout samples in regions similar to those in the pseudolabeled ROIs or is there a significant distribution shift?
> >
> >
> > We updated figures ‘Map of dataset’ in the main text and ‘Map of hydrological basins’ in the Datasheet to indicate ROI locations and whether or not ROI contain expert annotations. Overall, 13.4% of our dataset‘s contained ROI feature expert-labelled annotations. The distribution of ROI across partitions in the 5-fold cross-validation scheme is given below, where columns 2-4 report ROI-wise statistics and columns 5-6 outline tile-wise statistics.
> >
> >
> > | Partition | Total    | Generated | Kuro Siwo | Generated | Kuro Siwo |
> > |-----------|----------|-----------|-----------|-----------|-----------|
> > | 0         | 72       | 60        | 12        | 29097     | 2642      |
> > | 1         | 33       | 32        | 1         | 19224     | 505       |
> > | 2         | 69       | 63        | 6         | 28930     | 640       |
> > | 3         | 62       | 56        | 6         | 30712     | 5923      |
> > | 4         | 59       | 44        | 15        | 26827     | 5821      |
> >
> >
> > > It seems the purpose of Figure 5 is to justify that the pseudolabels (and expert annotations) generally match the distribution of floods expected from DFO. Differences from DFO are attributed to the near-coastal focus of this dataset. How else could differences arise, e.g., quality of pseudolabels?
> >
> > The message conveyed by Figure 4 is that the selected (near-)coastal ROI are representative by following DFO’s distribution to a reasonable extent. We hypothesise that differences between both distributions can be primarily attributed to our overall ROI selection strategy: As mentioned, a key role may be our core objective of focusing on (near-)coastal regions. Beyond that, the manner in which we avoid data overlap may cause differences by thinning out closeby ROI and event locations:
> > - When ROI candidates are incrementally selected for inclusion in the dataset, no ROI closer than circa 500 km to one another are considered. That is, a ROI is only included in the dataset if it is at least 500 km apart from the closest neighbouring ROI. This is to exclude any overlap of contextual information, as given by e.g. ERA5 & ERA5-Land.
> >
> > We suppose that the quality of generated labels has a comparably minor effect on these distributions. Label errors could cause distribution mismatch at the pixel-level that may even accumulate into the tile-level. In comparison, the aforementioned filtering criteria operate on the ROI- and event-level — which have a significant impact on the dataset statistics per each ROI or event decided to (not) be included. In return, this means that Figure 4 does not necessarily validate the generated labels but primarily makes a statement about the representativeness of the overall ROI and event selection strategy.
> >
> > > There are several flood detection datasets (e.g. Sen1Floods11, Sen12-FLOOD, etc). It would be helpful to have a more granular comparison of the differences between the GFF dataset and existing datasets (e.g., geographic coverage, number of samples, etc.)
> >
> >
> > Thanks, we appreciate the valuable suggestion! The table was as well requested in another review, so we inserted Table ‘Overview of datasets’ in the main text to provide the reader with a better overview of the most relevant prior work.
> >
> > > I did not see a maintenance plan for the dataset
> >
> > Thanks for the thoughtful inquiry! We included the following information in the Datasheet:
> >
> > *’ The dataset is hosted on Zenodo, which offers DOI for datasets and long-term availability. The code to generate and extend the dataset as well as to reproduce the benchmarks is hosted on GitHub. Both hosting solutions are established platforms and are expected to provide long-term maintenance. The authors will be providing support with the dataset through GitHub, e.g. resolving pull requests and open issues. Furthermore, a leaderboard of the best (peer-reviewed) models will be maintained in the repository’s README.’*
> >
> >
> > Beyond this, the authors may provide their best efforts to liase with rapid mapping experts or potential collaborators for follow-up. We can’t promise positive outcomes of these intents as of now, but circumstances that underline our expertise may facilitate these efforts.
> >
> > > In Table 1, W is used for water pixels on the left and F for flood pixels on the right. Are these meant to be the same?
> >
> > Yes, these are the same. Thanks for spotting this inconsistency, we harmonized both cases and updated Table ‘Track 1’ accordingly.
> >
> > [continuing in adjacent post]

---

> > > ### Author Rebuttal · Authors · 2024-08-18
> > >
> > > [continuing in adjacent post]
> > >
> > > > The paper states, "To our knowledge, there are no existing solutions processing such significant differences in scale" referring to ERA5 and S1. Presto includes ERA5 and S1: https://arxiv.org/abs/2304.14065
> > >
> > >
> > > Thanks for the valuable reference! Presto [ii] is a *pixel-based*, self-supervised model that uses masked autoencoding to pre-train a model on S1, ERA5 and a few more modalities. Although it does use S1 and ERA5, it does so in a *pixel-based* manner, which completely avoids the problems we were considering in this passage. We were thinking about the difficulties for spatial models (in this case CNNs) to handle *spatial maps* with such a large difference in scale. Presto doesn’t have to do anything special; their model only uses a single pixel from each modality. They never have to match up spatial 2D maps across these modalities. We have made this explicit in our revision: *’To our knowledge there are no existing solutions processing spatial information at such difference in scale. Although there exist works that utilise both S1 and ERA5 at the same time [69], they do not do so while modelling spatial relationships.’*
> > >
> > >
> > > Typical methods for spatial models would be to just resample one scale to the other at some point in the model architecture and concatenate along the channels. But, in this case, the difference of scale is so large that resampling becomes degenerate. The entire local patch fits within a single pixel of the context patch.
> > > So, one may wonder if a pixel-based method would be a neat way to avoid this issue. However, **we hypothesize flood forecasting to be an inherently spatial relationship**. Rain landing on distant areas, and the topography of the land are critical aspects of flood forecasting (see Caravan [iii]) which would not be modelled using a pixel-based approach.
> > >
> > >
> > > ---
> > >
> > > [i] Bountos, N. I., Sdraka, M., Zavras, A., Karasante, I., Karavias, A., Herekakis, T., ... & Papoutsis, I. (2023). Kuro Siwo: 12.1 billion $ m^ 2$ under the water. A global multi-temporal satellite dataset for rapid flood mapping. arXiv preprint arXiv:2311.12056.
> > >
> > > [ii] Tseng, G., Cartuyvels, R., Zvonkov, I., Purohit, M., Rolnick, D., & Kerner, H. (2023). Lightweight, pre-trained transformers for remote sensing timeseries. arXiv preprint arXiv:2304.14065.
> > >
> > > [iii] Kratzert, F., Nearing, G., Addor, N., Erickson, T., Gauch, M., Gilon, O., ... & Matias, Y. (2023). Caravan-A global community dataset for large-sample hydrology. Scientific Data, 10(1), 61.

---

> > > > ### Comment · Reviewer_ZKym · 2024-08-20
> > > > **A benchmark dataset on flooding in SAR data without any experts who can interpret flooding in SAR data?**
> > > >
> > > > I appreciate the authors providing detailed answers to my questions. However, the responses revealed something that I find very alarming:
> > > > "But, unfortunately, none of the authors are formally trained as rapid mapping experts to interpret SAR data for flooding. We reached out to a few such experts, but none responded."
> > > >
> > > > The dataset is for model prediction of flooding in SAR data, but has not been developed by or reviewed by anyone who is an expert in identifying flooding in SAR data. In my opinion, this is a major limitation of the dataset. SAR data is notoriously difficult to interpret and I don't see how visual review by untrained people (including "visitors") can be taken as confident verification of the dataset. This point exacerbates my concern about the effect of pseudolabels in the dataset.

---

> > ### Author Rebuttal · Authors · 2024-08-21
> >
> > ### Dear reviewer ZKym,
> >
> > Your doubts are a valuable viewpoint in this discussion. We appreciate the review for **encouraging us to provide empirical evidence** via  additional experiments and analysis in the [preceding rebuttal](https://openreview.net/forum?id=I2VOdtAc3H&noteId=O3pHhVBnSf) to demonstrate the quality of the provided labels. Hence **we kindly ask you to also take into account and comment on the empirical outcomes highlighted in the rebuttal**. Specifically,
> >
> > - our labels are generated by models accomplishing 0.88 & 0.87 F1 score for the minority class and > 0.99 F1 score for the majority class, **both on held-out expert-annotated labels**. This constitutes solid labelling quality and leaves plenty of room for improvement on our forecasting task.
> >
> > - we conducted an additional experiment (see Table *'Ablation studies by restricting training examples'* of the Supplementary Material) in response to the request to convince readers our masks are high quality. The outcomes show: even when **exclusively** evaluating on hold-out expert-labeled data, forecasting models **trained on *all* labels (expert-annotated + generated) perform considerably better than models trained on expert-annotated labels *only*.** This is evidence for the benefits of our generated labels.
> >
> > We wish to underline that we manually reviewed each generated mask for plausibility: No (co-)author is specifically trained in rapid mapping of SAR data — but we are by no means *’untrained’* either: N.L. and P.E. are experienced in the usage of SAR data (Sentinel-1, amongst several) for **monitoring** and SAR data (Sentinel-1) for **change detection**, respectively. K. D. is a hydrologist. Our external discussion partners involved further hydrologists and researchers working with Sentinel-1 measurements. We hope this partly eases your concerns.
> >
> > We consider that these are strong arguments in defence of our work and would value your critical thoughts on these. Thanks.
> >
> > Kind regards,
> >
> > GFF authors

---

### Official Review · Reviewer_R2fu · 2024-08-07
**A Novel Dataset and Benchmark for Prediction of Flood Inundation**

**Rating:** 8
**Confidence:** 3
**Correctness:** Yes, all claims appear to be correct.
**Clarity:** The paper is well-structured and easy…

**Review:**

**Strengths**
- The paper is well-structured and easy to follow, providing a clear background on flood prediction and highlighting the importance of creating this benchmarking dataset.
- The authors provide detailed instructions on building the dataset and explain their reasoning behind the decisions made.
- The work includes a diverse range of baselines and presents a comprehensive empirical study, demonstrating the practical applications of the dataset.
- The authors offer constructive insights for future research in this domain.

**Weaknesses**
- The experiments rely solely on the F1 metric and its variants, which may limit the evaluation of the dataset's performance. Adding more evaluation metrics could strengthen the results.

**Strengths:**

See **'Strengths'** above.

**Additional Feedback:**

N/A

**Documentation:**

The paper itself and its supplement has already provided sufficient details. Additionally, both the code and data are publicly accessible.

**Ethics:**

I have no specific ethical concerns.

**Limitations:**

Yes, the authors have thoroughly addressed future work directions in a dedicated section.

**Opportunities For Improvement:**

See **‘Weaknesses’** above.

**Relation To Prior Work:**

The paper clearly highlighted the difference between this work and the existing papers.

**Summary And Contributions:**

This paper presents a new dataset and benchmark for predicting flood extent, which is critical for mitigating the devastating effects of floods. The dataset links weather prediction and spaceborne flood mapping, enabling the direct forecasting of flood extent. The benchmark provides a comprehensive platform for evaluating flood forecasts, with two series for general and coastal regions respectively.

---

> ### Author Rebuttal · Authors · 2024-08-18
>
> ## Dear reviewer R2fu,
>
> Thanks once more for your constructive feedback, your peer review and suggestions helped us improve on our initial submission. In the following, please find our rebuttal to your review and please also have a look at the main post outlining all further improvements. We hope our replies to your suggestions confirm your positive sentiment.
>
> > The experiments rely solely on the F1 metric and its variants, which may limit the evaluation of the dataset's performance. Adding more evaluation metrics could strengthen the results.
>
>
> Thanks for the valuable suggestion. We included Table ‘Extended metrics’ in the Supplementary Material, following the Datasheet. The table complements the main findings with additional metrics of Intersection over Union (IoU) and Accuracy. For your convenience, you may also find the table below.
>
> **Table:** *Extended metrics for all five baselines, evaluated on every ROI. Metrics tend to agree besides minor differences, with U-TAE performing best.*
>
> | Model               | F1                  | F1-B                | F1-W                | IoU                 | IoU-B               | IoU-W               | Accuracy            | Accuracy-B          | Accuracy-W          |
> | ------------------- | ------------------- | ------------------- | ------------------- | ------------------- | ------------------- | ------------------- | ------------------- | ------------------- | ------------------- |
> | **U-TAE**           | **0.77 ± 0.04**     | **0.97 ± 0.00**     | **0.57 ± 0.07**     | **0.67 ± 0.04**     | **0.94 ± 0.01**     | **0.40 ± 0.07**     | **0.81 ± 0.06**     | **0.96 ± 0.01**     | 0.65 ± 0.14         |
> | **LSTM U-Net**      | 0.76 ± 0.04         | **0.97 ± 0.00**     | 0.55 ± 0.08         | 0.66 ± 0.04         | **0.94 ± 0.01**     | 0.39 ± 0.07         | 0.80 ± 0.07         | **0.96 ± 0.01**     | 0.65 ± 0.14         |
> | **3DConv U-Net**    | 0.76 ± 0.04         | 0.97 ± 0.01         | 0.54 ± 0.08         | 0.66 ± 0.04         | **0.94 ± 0.01**     | 0.38 ± 0.07         | 0.79 ± 0.07         | 0.96 ± 0.02         | 0.63 ± 0.16         |
> | **MaxViT U-Net**    | 0.75 ± 0.03         | 0.96 ± 0.01         | 0.53 ± 0.06         | 0.65 ± 0.04         | 0.93 ± 0.02         | 0.37 ± 0.06         | 0.80 ± 0.05         | 0.95 ± 0.03         | 0.65 ± 0.12         |
> | **logistic regression** | 0.66 ± 0.04     | 0.93 ± 0.02         | 0.40 ± 0.07         | 0.56 ± 0.04         | 0.87 ± 0.02         | 0.26 ± 0.06         | 0.80 ± 0.06         | 0.88 ± 0.03         | **0.72 ± 0.13**     |
>
>
> *To complement the baseline benchmarks in the main text with additional analysis, the F1 scores are complemented with metrics of Intersection over Union (IoU) and Accuracy. All metrics are provided as overall scores, as well as specifically evaluated on the background (B) and water (W) classes.*

---

### Official Review · Reviewer_CFWK · 2024-08-09
**Off to new Shores**

**Rating:** 7
**Confidence:** 3
**Correctness:** It appears correct.

**Review:**

The authors used an earlier version of Kuro Siwo [not the version submitted to Neurips] as validation of after-flood to assess the accuracy of their predictions. It includes 298 Regions of Interest (ROI). The dataset comprises flood drivers as input and flood segmentation maps at event time as targets.
Drivers include Sentinel 1 data triplets, 2 images pre-flood and one at the time of flooding, DEM, HAND, ERA5 and ERA5-Land for hydrometeorological conditions, HydroATLAS for hydro environmental conditions.

**Strengths:**

It's nice to see a paper on flood forecasting rather than flood mapping, which is what most papers focus on.

**Additional Feedback:**

NA

**Clarity:**

The paper is well written, but some of the reasons behind using those variables as predictors of floods would be useful. For example, a table that cites other works that were found important. Also, why were climate variables such as precipitation and temperature used?

**Documentation:**

GitHub is well documented

**Ethics:**

Seem fine

**Limitations:**

See Qs above

**Opportunities For Improvement:**

Pls see Qs above

**Relation To Prior Work:**

Yes

**Summary And Contributions:**

The paper presents Global Flood Forecasting, a newly curated dataset enabling flood extent forecasting, and focuses on two benchmarks, one in general and one in coastal areas.

The flood events come from the Dartmouth Flood Observatory, which is incomplete and has several issues. Have the authors considered augmenting this dataset?

How was the map of flood hazard frequency in Figure 2 produced? I see a study cited from 2005 that's pretty old. Would you now expect that the flooding frequency has changed in the past 20 years?

Why is ESA WorldCover used for permanent water? Have the authors considered dynamic permanent water layers such as the one by Pekel et al.?

Have the authors used lagged variables?

---

> ### Author Rebuttal · Authors · 2024-08-18
>
> ## Dear reviewer CFWK,
>
> Thanks once more for your constructive feedback, your peer review and suggestions helped us improve on our initial submission. In the following, please find our rebuttal to your review and please also have a look at the main post outlining all further improvements. We hope our replies to your questions confirm your positive sentiment and may clear any remaining concerns.
>
> > The flood events come from the Dartmouth Flood Observatory, which is incomplete and has several issues. Have the authors considered augmenting this dataset?
>
> The Dartmouth Flood Observatory (DFO) catalogue is commonly acknowledged in the literature for being amongst, if not the most complete resource of flood events [i, ii]. Yet we agree with the reviewer that there’s still plenty of opportunity for improvements. In particular, our experience with DFO is that it may benefit from further refinements in terms of the spatial regions and temporal windows associated with reported flood events, for which we had to compensate via our ROI selection process.
>
> A contribution of our submission is in **diversifying the ROIs** selected according to DFO’s impact metrics with a curated collection of 280 tropical cyclone landfall ROI and subsequent floods identified in [iii]. Please see sections “3.1 Identifying candidate Regions Of Interest” of the main text for detailed information.
> 1. this diversified the ROIs selected according to our impact metrics and
> 2. the cause of flood events in DFO is oftentimes not specified in detail, this additional data source provided more information.
>
> In fact, we think that a significant direction for improving over the DFO catalogue would be refining its data of flood causes and drivers. We welcome suggestions for further data sources of potential flooding events.
>
> > How was the map of flood hazard frequency in Figure 2 produced? I see a study cited from 2005 that's pretty old. Would you now expect that the flooding frequency has changed in the past 20 years?
>
> The referenced flood hazard frequency information a product provided in World Bank’s primer on ‘Natural disaster hotspots: a global risk analysis’, distributed as an analysis-ready layer via [NASA’s SEDA platform](https://sedac.ciesin.columbia.edu/data/set/ndh-flood-hazard-frequency-distribution).
>
> The World Bank product was **derived from DFO records in 1985 - 2003.** That is, the analysis-ready product does not include the most recent 20 years. Unfortunately, we don’t have access to the algorithm with which the original layer was derived and thus would have to produce our own product. However, the scientific consensus is that **climate change may bring a sharp increase in (near-)coastal hazards**, e.g. due to rise in mean sea levels [iv, v] and more intense extreme storm events [vi, vii]. Therefore, we hypothesise that (near-)coastal sites as covered by our work may experience **increasing hazard risk and thus remain relevant for future relief efforts.**
>
> > Why is ESA WorldCover used for permanent water? Have the authors considered dynamic permanent water layers such as the one by Pekel et al.?
>
> ESA WorldCover is used to complement generated maps with permanent water labels because it provides information of areas that are generally under water, year-to-year. Primarily, the product masks lakes, large rivers and the ocean from the “water” predictions, providing a better estimate of flood prediction. We agree with the reviewer that **not capturing inter- and intra-annual surface water changes is a limitation.** However, having considered alternative solutions earlier plus the referenced product in greater detail, we concluded that either alternative involves a trade-off and thus argue that the **usage of ESA WorldCover is nonetheless reasonable.**
>
> We previously considered alternatives like DynamicWorld, but we didn’t explore GSWE to a sufficient extent, so we thank the reviewer for the suggestion. Initially, we were hopeful that it could improve our work, but eventually concluded that ESA WorldCover is roughly equivalent for our purposes, and operates at a better resolution. We apologise for the large response; it’s a complex point, and deserves a comprehensive answer.
>
> ### Theoretical
>
> Using a multitemporal water map puts the burden of defining “permanent water” onto us. The dichotomy between “permanent water” and “flood water” is just a useful fiction. In reality, there is a spectrum between permanent and temporary water, and we felt that creating a new way to separate these extremes is somewhat beyond our scope. We have an intuition that we want our “permanent water” to include very permanent water from lakes, large rivers and ocean areas, but not necessarily large, seasonally flooded regions. After all, if the Kuro Siwo models correctly identify that there is now water where there wasn’t before, our dataset shouldn’t care whether that happened last year as well. On this spectrum, we believe ESA WorldCover is biased towards only reporting such very permanent water as “permanent water”. So, currently, we leverage ESA WorldCover’s definition as a reasonable guess.
>
> ### Practical
>
> If we were to use GSWE instead, we would use their undocumented YearlyClassification dataset to define “permanent water” (see below). This data has 3 different water classes, which appear to be increasing degrees of permanence (classes 1-3). We visualised ESA WorldCover and GSWE’s YearlyClassification in two locations to help make these decisions.
>
> We noticed that the most strict definition from GSWE (class 3) generally covered slightly less area than ESA WorldCover. Thus, if we use GSWE class 3 only, it would be equivalent to using a slightly stricter definition of “permanent”. Practically, this would mean we evaluate on more pixels that are “usually under water, but not always” using GSWE class 3.
>
> [continuing in adjacent post]

---

> > ### Author Rebuttal · Authors · 2024-08-18
> >
> > [continuing in adjacent post]
> >
> >
> > It seems to us, then, that using GSWE’s YearlyClassification class 3 is the best choice from GSWE. But this is roughly the same as ESA WorldCover’s “permanent water”, and at a coarser resolution (30 m instead of 10 m) besides. So, we argue, we should continue to use ESA WorldCover.
> >
> > ### Why Yearly Classification?
> >
> > There are several different layers available from GSWE. There are high quality aggregated maps: Occurrence, Change, Seasonality, Recurrence, Transitions and Maximum extent. However, these aggregated maps are only available for 2021, currently. Thus, using these sources would be similarly monotemporal as ESA WorldCover, but at a coarser resolution.
> >
> > On their FTP download server there are other, undocumented options to download, including MonthlyHistory and YearlyClassification datasets. Upon inspection, the MonthlyHistory maps are not fit for purpose, including many obvious errors. The YearlyClassification images appear to be sound, and include three different water levels.
> >
> > > Have the authors used lagged variables?
> >
> > We have **not explicitly used lagged variables as of now**, but this may be an interesting direction for future modelling ventures.
> >
> > The reason we have not explicitly modelled lagged variables is because **event dynamics may implicitly be learned** by our current baselines. The dataset includes both pluvial and fluvial events. For the former, we expect the primary relationship between rain and flooding to appear within a short time period. For the latter, there may be lagged effects from rainfall up to weeks after the rainfall. Thus, we choose to include atmospheric variables in a window of 20 days to capture the vast majority of both immediate and lagged effects. We believe that deep learning models which have access to both rainfall and a DEM or HAND, can, theoretically, learn to somewhat integrate this information and automatically discover such lagged effects where they exist. Thus we consider our baselines sufficient without explicitly including a lagged variable, but this may prove beneficial in further research.
> >
> > > The paper is well written, but some of the reasons behind using those variables as predictors of floods would be useful. For example, a table that cites other works that were found important. Also, why were climate variables such as precipitation and temperature used?
> >
> > Thanks for your kind words and the valuable feedback! An overview table of prior work was also requested by another reviewer. Following your request, we **inserted Table ‘Overview of datasets’ in the main text** (see below) to provide the reader with a better overview of relevant prior work. With the insertion of the GloFAS streamflow & runoff data product we also inserted a novel paragraph to provide more literature background in section ‘2 Related Work 2.1 River streamflow & runoff forecasting’.
> >
> > | **Dataset**                                             | **Task**         | **Sample size**    | **Resolution**           | **Sample count** | **Static input**               | **Dynamic input**                             | **Event count** | **Timestamps**      |
> > | ------------------------------------------------------- | ---------------- | ------------------ | ------------------------ | ---------------- | ----------------------------- | -------------------------------------------- | --------------- | ------------------- |
> > | SEN12-FLOOD | classification   | 512 × 512          | 10 m                     | 336              | -                             | S1, S2                                       | 3               | circa 9-14          |
> > | OMBRIA | segmentation     | 256 × 256          | 10 m                     | 1,688            | -                             | S1, S2                                       | 23              | 1 Pre + Post        |
> > | S1GFloods | segmentation     | 256 × 256          | 10 m                     | 5,360            | -                             | S1                                           | 46              | 1 Pre + Post        |
> > | CAU-Flood | segmentation     | 256 × 256          | 10 m                     | 18,302           | -                             | S1, S2                                       | 18              | 1 Pre + Post        |
> > | Kuro Siwo | segmentation     | 224 × 224          | 10 m                     | 67,490           | DEM                           | S1                                           | 43              | 2 Pre + Post        |
> > | GRDC GRDB | regression       | 1D sequence        | in-situ                  | 10,000+          | -                             | river gauges                                  | -               | 10,000+             |
> > | HYSETS | regression       | 1D sequence        | in-situ, basin & 10-30 km | 14,425           | basin properties              | river gauges, NRCan + SCDNA + Livneh + ERA5  | -               | 10,000+             |
> > | Caravan | regression       | 1D sequence        | in-situ & basin          | 10,000+          | HydroATLAS                    | river gauges, ERA5-Land                      | -               | 10,000+             |
> > | **Global Flood Forecasting (ours)**                     | **segmentation** | **224 × 224**      | **10 m & 5-30 km**       | **163,873**      | **DEM, HAND, HydroATLAS**     | **S1, GloFAS, ERA5(-Land)**                  | **298**         | **20 Pre + Post**   |
> >
> > **Table:** *Overview of datasets for flood mapping (top) and flood forecasting (bottom) purposes. The former features high-resolution imaging, while the latter focus on in-situ time series. GFF enables forecasting of flood extent by curating sequences of gridded products at high spatial resolution.*
> >
> > [continuing in adjacent post]

---

> > > ### Author Rebuttal · Authors · 2024-08-18
> > >
> > > [continuing in adjacent post]
> > >
> > > ---
> > >
> > > [i] https://global-flood-database.cloudtostreet.ai/
> > >
> > > [ii] Tellman, B., Sullivan, J. A., Kuhn, C., Kettner, A. J., Doyle, C. S., Brakenridge, G. R., ... & Slayback, D. A. (2021). Satellite imaging reveals increased proportion of population exposed to floods. Nature, 596(7870), 80-86.
> > >
> > > [iii] Titley, H. A., Cloke, H. L., Harrigan, S., Pappenberger, F., Prudhomme, C., Robbins, J. C., ... & Zsótér, E. (2021). Key factors influencing the severity of fluvial flood hazard from tropical cyclones. Journal of Hydrometeorology, 22(7), 1801-1817.
> > >
> > > [iv] Taherkhani, M., Vitousek, S., Barnard, P. L., Frazer, N., Anderson, T. R., & Fletcher, C. H. (2020). Sea-level rise exponentially increases coastal flood frequency. Scientific reports, 10(1), 1-17.
> > >
> > > [v] Kirezci, E., Young, I. R., Ranasinghe, R., Muis, S., Nicholls, R. J., Lincke, D., & Hinkel, J. (2020). Projections of global-scale extreme sea levels and resulting episodic coastal flooding over the 21st Century. Scientific reports, 10(1), 1-12.
> > >
> > > [vi] Bevacqua, E., Vousdoukas, M. I., Zappa, G., Hodges, K., Shepherd, T. G., Maraun, D., ... & Feyen, L. (2020). More meteorological events that drive compound coastal flooding are projected under climate change. Communications Earth & Environment, 1(1), 47.
> > >
> > > [vii] Gori, A., Lin, N., Xi, D., & Emanuel, K. (2022). Tropical cyclone climatology change greatly exacerbates US extreme rainfall–surge hazard. Nature Climate Change, 12(2), 171-178.

---

### Author Rebuttal · Authors · 2024-08-11

### Dear reviewers,

We are **thankful for the most constructive feedback, questions and comments.**
Overall, the submission has been received positively: The reviews highlighted the relevance of the topic and the literature gap closed by our work (reviewer CFWK & vRcU), appreciated the submission to *"address an important task"* and being *"thoughtfully designed"* (reviewer ZKym) while acknowledging it as *"technically sound and of high quality"* (reviewer vRcU). On top, we are glad for the valuable questions and guidance, which encourage us to provide clarifications and revise the submission to provide a better version of our work.
Thanks a lot, we appreciate the efforts put into your reviews!

The recommended rebuttal date is August 16th. However, with the venue's initial delay of review releases we may as well require 2-3 additional days of your patience.


We are **positive to address the raised questions, concerns and suggestions to a satisfying extent** and look forward to an engaging rebuttal period!

Kind regards,

GFF authors

---

### Author Rebuttal · Authors · 2024-08-18

### Dear reviewers,


please find our revised **main text** here.



Kind regards,

GFF authors

---

### Author Rebuttal · Authors · 2024-08-18

### Dear reviewers,


please find our revised **Datasheet & Supplementary Material** here.



Kind regards,

GFF authors

---

> ### Author Rebuttal · Authors · 2024-08-26
>
> ### Dear reviewers,
>
> please find our updated **Datasheet & Supplementary Material** here, now including ablations on the input modalities.
>
>
> Kind regards,
>
> GFF authors

---

### Author Rebuttal · Authors · 2024-08-18

## Dear reviewers,


**Thanks once more for the constructive and overall positive feedback**. Complementing the more detailed replies to each individual review, in this post we wish to highlight our submission’s main improvements following your suggestions and beyond.


### Overall consensus

Altogether, our submission was received well, and most feedback were minor suggestions to further improve upon the initial submission. These primarily included questions or requests for clarifications, to which we replied in text. We particularly valued those of your requests which motivated us to provide additional content, either in the main text or in the supplementary material. We hope these adjustments are in the spirit of your requests and will confirm your positive sentiment or resolve any remaining concern.

We uploaded the **updated versions of the main text and the supplementary material** with **changes highlighted** in colored text, to facilitate the review process. Upon acceptance, we will upload the camera-ready version without this highlighting.


### Inclusion of new data modality: River streamflow & runoff reanalysis

The initial submission featured all code all to replicate the dataset curation and reported outcomes. In addition, we underlined the provided code’s value to extend the dataset to new scenarios or modalities. As part of the rebuttal, we demonstrate this value by **including novel data modality provided by ECMWF’s Global Flood Awareness System (GloFAS).** The new data includes daily aggregated river streamflow and runoff reanalysis with 0.05 degrees resolution at the context scale. Accordingly, we expanded the Related Work section to include a subsection on Streamflow forecasting and included a paragraph on GloFas in section 3.2.

Finally, we probed the relevance of this modality by re-training the best-performing model previously identified by inserting GloFAS data as an additional forcing at the context level. The outcomes are included in Tables ‘Track 1’ and ‘Track 2’ of the main text. Notably, we don’t see clear improvements by including the new GloFAS data, with the exception of ROI which are more inland and further away from coastlines. This may be due to established river streamflow and runoff products being optimized for non-coastal areas (e.g. GloFAS is [only calibrated further inland](https://www.un-spider.org/sites/default/files/20230220_1_glofas_intro_products_overview.pdf#page=9) ), which is an exciting direction to further investigate and was amongst the reasons to develop our dataset and benchmark.


### New baseline: simple logistic regression model

Following the prior work of Gerard et al (2023), we included a **simple logistic regression model as a fifth baseline**. The main benefit of this model is its simplicity and interpretability. Accordingly, we introspect and **interpret its learned weights as an indicator of feature importance**. The model’s performance is included in Tables ‘Track 1’ and ‘Track 2’ of the main text, and the feature weights are summarised in Table ‘Importance analysis’ of the Supplementary Material.


### Ablations experiments

Following the request of a review to underline the quality of the generated labels, we **re-trained our best performing model on ablations of ROI**. The setup has the model trained on data and target labels which are
1. Generated-only
2. Only Expert-annotated
3. All data

The results are reported in Table ‘Ablation studies’ of the Supplementary Material. Predictions are analysed on all ROI as well as on hand-annotated ROI only. The outcomes show that the model performs best when utilizing all data, generated and hand-labelled alike, which provides further evidence of the quality of the generated labels.

Furthermore, we are currently running **ablation experiments on the provided modalities**. While these were *not* requested by any reviewer, we consider they **may provide valuable insights for the reader** and future modelling efforts. Currently, our machines are occupied with these runs and we foresee to provide outcomes by the end of next week. We expect no issues and will insert the ablation outcomes in a new Table in the Supplementary Material.


### Evaluation on extended metrics

Following the request of a review to complement the main findings with **additional measures of performance**, we present Table ‘Extended metrics’ in the Supplementary Material. The additional table complements the evaluation of the baseline models via F1 score with the additional metrics of IoU and Accuracy. We believe this will provide valuable diagnostics for the reader.

[continuing in adjacent post]

---

> ### Author Rebuttal · Authors · 2024-08-18
>
> [continuing in adjacent post]
>
>
> ### Other improvements and remarks
>
> - Following the request by two reviewers, we inserted Table ‘Overview of datasets’ into the main text to provide the reader with information of related work at a glance.
> - We identified and resolved a bug that affected the initialization of U-TAE’s affine modulation. The issue was specific to U-TAE and did not affect any other model, but improved U-TAE’s performance notably.
> - Following profiling of the provided code, we identified that reading ERA5 & ERA5-Land data constituted a computational bottleneck. Specifically, we experienced that loading TIF files was substantially slower than reading the same data when provided as NETCDF file. Accordingly, ERA5 & ERA5-Land are now distributed in the more efficient NETCDF file, which provides notable speedup.
> - Following introduction of the GloFAS streamflow & river runoff reanalysis product provided at 0.05 degrees vertical resolution, we resampled and provided all other context scale data at the same spatial resolution.
> - We implemented a novel feature, which explicitly derives a binary land-sea mask from the DEM at context and local scale, to provide models with an auxiliary input feature.
> - Following the aforementioned changes, we re-ran all experiments and report the updated outcomes in the revised documents. Compared to the initially reported numbers, we observed improved performance and a new best-performing model, achieving an overall F1 score of about 0.77.
>
>
> Overall, we believe that these adjustments improve the quality of our initial submission. The changes were driven by our intrinsic aim to provide further refinements as well as **your thorough reviews and suggestions, for which we are most thankful**. We hope that you are considering this a strong rebuttal, and that our efforts may reinforce your positive sentiment.
>
>
> Kind regards,
>
> GFF authors

---

> > ### Author Rebuttal · Authors · 2024-08-26
> >
> > ### Dear reviewers,
> >
> > As announced above, we provide an additional update to the **Datasheet & Supplementary Material**, now featuring **ablation experiments on the provided modalities**.  While these were *not* requested by any reviewer, we consider they **may provide valuable insights for the reader** and future modelling efforts. You can find the novel experiment outcomes at the end of the Supplementary Material (see post below).
> >
> > Kind regards,
> >
> > GFF authors

---

### Decision · Program_Chairs · 2024-09-26

**Decision:**

Accept (Poster)

**Comment:**

The ratings for this paper are pretty high, but the confidence levels are low except for two reviews (one very positive and the other that is not entirely positive). As someone who works in a related area, I am concerned by the statement about quality control raised by the latter reviewer. Relatedly, there are well-known datasets that the remote sensing community uses for evaluating flood mapping techniques from SAR images (for example, the carefully curated Houston dataset (Harvey floods from 2016?) has been used extensively in the community - see Liang et al. 2018, SDM).

 If Neurips offers the option of shepherded acceptance, this paper could be a candidate. However, if it does not, consider this issue carefully before accepting.

Given the scores - I am recommending acceptance but will leave it to the PC chairs to decide on this issue (which is essential).